# Acceleration of the ocean warming from 1961 to 2022 unveiled by large-ensemble reanalyses

Andrea Storto [1] ✉ & Chunxue Yang[1]

Long-term changes in ocean heat content (OHC) represent a fundamental global warming indicator and are mostly caused by anthropogenic climate-altering gas emissions. OHC increases heavily threaten the marine environment, therefore, reconstructing OHC before the well-instrumented period (i.e., before the Argo floats deployment in the mid-2000s) is crucial to understanding the multi-decadal climate change in the ocean. Here, we shed light on ocean warming and its uncertainty for the 1961-2022 period through a large ensemble reanalysis system that spans the major sources of uncertainties. Results indicate a 62-year warming of $0.43 \pm 0.08 \, \text{W m}^{-2}$, and a statistically significant acceleration rate equal to $0.15 \pm 0.04 \, \text{W m}^{-2} \, \text{dec}^{-1}$, locally peaking at high latitudes. The 11.6% of the global ocean area reaches the maximum yearly OHC in 2022, almost doubling any previous year. At the regional scale, major OHC uncertainty is found in the Tropics; at the global scale, the uncertainty represents about 40% and 15% of the OHC variability, respectively before and after the mid-2000s. The uncertainty of regional trends is mostly affected by observation calibration (especially at high latitudes), and sea surface temperature data uncertainty (especially at low latitudes).

Ocean warming is among the most notable threats to the marine environment, leading to e.g. sea-ice decline[1,2], sea-level rise[3], and an increase in frequency and amplitude of marine heatwaves[4], which, together with other climate change effects (e.g., ocean acidification[5]), all endanger the marine ecosystem[6]. The ocean is characterized by non-uniform warming[7], due, in turn, to the spatially and temporally varying heat uptake from the atmosphere and surface wind variability[8], combined with multi-scale redistribution through advective processes and vertical mixing. Ocean heat content (OHC, that is, the total amount of heat stored in the ocean) is one of the most important indicators of climate change in the ocean. While inter-annual variations of OHC are shaped by both internal climate variability and external forcing[9], its long-term increase is due to the human-induced increase of climate-altering gas concentrations[10,11]. Reliably quantifying ocean warming and its uncertainty has enormous importance for monitoring Earth's energy budget and climate[12,13].

Accounting for the main sources of uncertainty of OHC in the pre-Argo period is crucial to provide estimates with reliable uncertainty envelopes[14]. Before the Argo era (namely before the 2000s), uncertainty is dominated by the observations themselves, mostly the eXpendable BathyThermograph data[15,16]; however, horizontal mapping methods[17,18] and vertical interpolation (i.e., the error length-scales[19]) are all non-negligible sources of errors[20]. Further to these, the accuracy of model-based reconstructions may also be affected by the vertical physics parametrization uncertainty and systematic model errors[21], assumptions in the data assimilation systems, and the uncertainty in all other input datasets (notably, the atmospheric forcing and sea surface data[22,23]). Intrinsic (chaotic) ocean variability is also non-negligible for regional estimates[24], making the overall picture quite complicated.

The analysis of ocean heat content is usually performed through (i) objective analyses[17,25,26] which are statistical mapping of the

[1]Institute of Marine Sciences (ISMAR), National Research Council (CNR), Rome, Italy. ✉e-mail: andrea.storto@cnr.it

observations, (ii) reanalyses that use an ocean general circulation model to project forward in time the ocean state, using observations to constrain the model trajectory through data assimilation[27], (iii) a combination of them, for instance, using reanalyses to derive error statistics for unsampled regions[28], (iv) use of proxy data[29] or indirect data that infer OHC increase from satellite-derived steric sea level variations (i.e., the so-called geodetic estimates[14]), or (v) advanced data-driven techniques such as machine learning[30]. Reanalyses, while being very costly from a computational point of view, permit a multi-variate four-dimensional characterization of the ocean, useful in turn for process-oriented and cause–effect studies[31]; reanalyses are attractive because they can in principle ingest any measurement which is related to the ocean state (for instance, altimetry and gravimetry observations[32,33]), and for a much broader range of applications than objective analyses, notably to initialize long-range prediction systems[34].

The present study aims to re-assess the OHC trends and their uncertainty as estimated by reanalyses, compare them with objective analyses, and draft a hierarchy of sources of uncertainty. It builds upon a large-ensemble reanalysis system (32 members) that includes, in the ensemble generation, the major sources of uncertainty. We also take advantage of an objective analysis system (i.e., a statistical mapping of the observations using the same analysis system as the reanalysis but without any numerical ocean model integration), which is used to test if any of the reanalysis signals are spuriously given by the interaction between the ocean model and the data assimilation[35,36]. The goal is thus to quantify the warming and assess and rank the sources of uncertainty in the OHC reconstruction.

## Results

We summarize, in the next two sections, the main outcomes of the large ensemble reanalysis system (hereafter CIGAR, the Cnr Ismar Global historicAl Reanalysis, http://cigar.ismar.cnr.it) in terms of ocean warming distribution and uncertainties, respectively. For some uncertainty diagnostics, we divide the reanalysis period (1961–2022) into the observation-poor period 1961–2001 and the observation-rich period 2002–2022, considering that the Argo float deployment[37] starts in the early 2000s and matures around the mid-2000s.

### Ocean warming distribution

The total ocean warming from CIGAR in terms of energy per Earth's area is equal to $0.43 \pm 0.08$ W m⁻² for the period 1961–2022 (Table 1), and it equals $0.41 \pm 0.09$ W m⁻² for the period 1961–2020 coinciding with the latest GCOS assessment[38], which indicates $0.41 \pm 0.10$ W m⁻² for the same period. Note that we have followed the uncertainty definition of the GCOS assessment: assuming Gaussian distributions for the trend errors, the uncertainty is given as twice the ensemble standard deviation, which corresponds to the 95% confidence level[38]). Thus, the two estimates are identical in terms of trends, and very close in terms of uncertainty, confirming the two approaches are consistent and complementary, as expected. We found a significant global warming acceleration equal to $0.15 \pm 0.04$ W m⁻² dec⁻¹ for the 1961–2022 period. During the recent period 2006–2018, the acceleration is equal to $0.20 \pm 0.07$ W m⁻² dec⁻¹, in agreement within the error bars with the estimates $0.50 \pm 0.47$ W m⁻² dec⁻¹ over mid-2005 to mid-2019[39] and $0.25$ W m⁻² dec⁻¹ over 2002–2019[13], although CIGAR shows much smaller uncertainty.

Timeseries for the global ocean heat content anomalies are shown in Fig. 1a, together with the GCOS20 assessment[26]. Additionally, we display estimates from ZANNA19[40], which reconstructs OHC from sea surface data using Green's functions, and from ARANN[41], which uses autoregressive artificial neural networks. The figure shows that, although the full-period trend is similar, our reanalysis system shows enhanced interannual variability, and a pronounced non-linear OHC increase, with the first period (up to about 1997–2000) exhibiting a slower increase and a sharper increase afterward. The other timeseries

**Table 1 | Global warming (in W m⁻², in units of the Earth's surface) from CIGAR and other assessments**

| Dataset | Global warming (W m⁻²) 1961–2022 | Global warming (W m⁻²) 1961–2020 | |
|---|---|---|---|
| CIGAR | $0.43 \pm 0.08$ | $0.41 \pm 0.09$ | |
| GCOS22 | NA | $0.41 \pm 0.10$ | |
| **Dataset** | **OHC trend (W m⁻²) (1961–2018)** | **Interannual variability (1E9 J m⁻²) (1961–2018)** | **Acceleration (W m⁻² dec⁻¹) (1961–2018)** |
| CIGAR | 0.42 | 0.20 | 0.13 |
| GCOS20 | 0.34 | 0.09 | 0.07 |
| ARANN | 0.29 | 0.13 | 0.11 |
| ZANNA19 | 0.35 | 0.05 | 0.03 |
| OA | 0.41 | 0.19 | 0.12 |
| OA-BGSIM | 0.44 | 0.07 | 0.04 |
| OA-MON | 0.36 | 0.12 | 0.07 |
| OA-SLS | 0.37 | 0.14 | 0.09 |

To obtain the trend in units of the ocean surface, values need to be multiplied by 1.42. In both products, the uncertainty is defined as twice the ensemble standard deviations. The table also shows the trend, interannual variability, and acceleration of CIGAR, the GCOS20 assessment[26], ARANN[41] and ZANNA19[40], and several objective analysis experiments explained in the text, over the common period 1961–2018. Trends are computed as the slope of the fitted line (in units of the Earth's surface); interannual variability as the standard deviation of the detrended yearly means; acceleration from the coefficient of the fitted parabola. OA is the objective analysis experiment with the same setting as CIGAR but no ocean model and background defined as the 10-day climatology plus the persistent anomaly from the previous analysis cycle; OA-BGSIM is as OA but with the background from model simulations without data assimilation; OA-MON is as OA, but with 1-month assimilation frequency and time-window; OA-SLS is as OA, but with halved horizontal correlation length-scales.

show a steadier increase (see also Table 1 for quantitative diagnostics), similar also to CMIP simulations[42]. For the well-observed period 2007–2022, we also compare the reanalysis with other independent estimates, i.e., geodetic estimates[43] and CERES-based timeseries[28,44]. This comparison, in terms of OHC yearly tendencies, shows a very good agreement between the datasets, except for some well-known differences in the CERES-based timeseries (e.g., all datasets but those based on CERES exhibit cooling in 2016[45]). The high-frequency variability between CERES-based estimates and those from oceanic observations is known to be largely different, due to several weather and climate processes providing a different response in the TOA EEI (top-of-atmosphere Earth Energy Imbalance) compared to the ocean heat uptake[12]. The climate community is converging toward comparing these two complementary datasets at frequencies slower than 3 years[28,43,46]. However, an accurate understanding of the scale coherence between the two is still an open question, whose answer is complicated by the relatively short temporal record of the satellite data.

To gain confidence in the datasets, it is important to understand whether the enhanced interannual variations in the reanalysis system are somehow spurious; alternatively, the objective analysis methodology and the ensemble average operation performed over the quite diverse range of products of the GCOS assessment may flatten the ocean heat content tendency signals. To this end, we compare our ensemble mean realization with that of a corresponding objective analysis (same assimilation system but no ocean dynamical model, see "Methods" section for details), denoted OA in Fig. 1 and Table 1. This objective analysis has a trend, acceleration, and interannual variability very close to that of CIGAR. In one alternative objective analysis experiment, the background—namely, the prior estimate used in the statistical analysis—is taken from an external model simulation like CMIP (OA-BGSIM), mimicking to some extent OA methodologies relying on external model-based products as background[47]. This experiment shows a much more attenuated variability and a steady

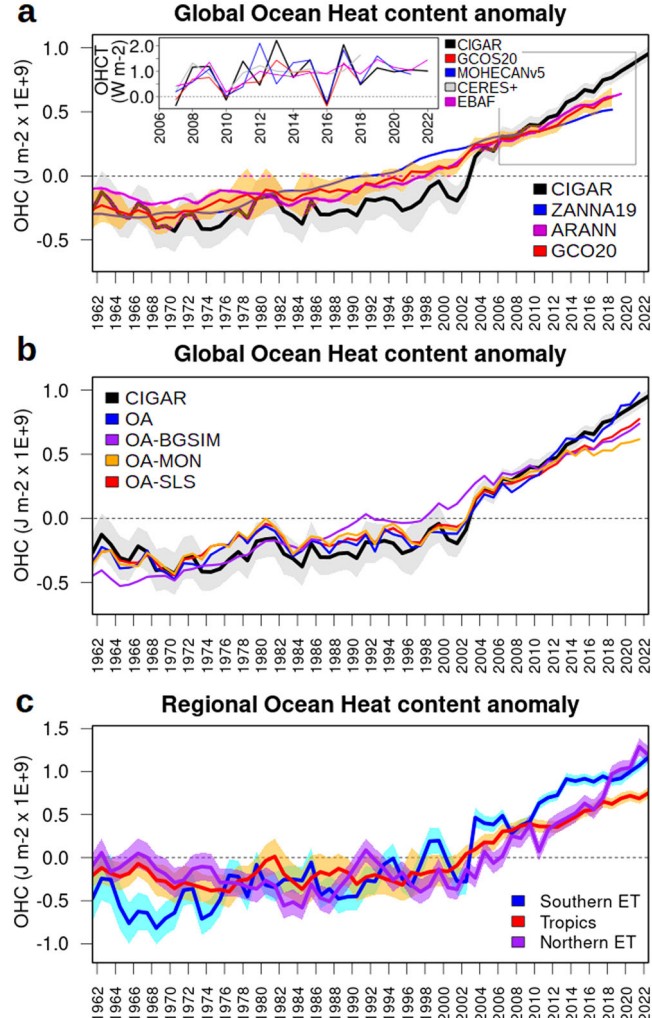

Fig. 1 | **Ocean heat content evolution. a** Ocean heat content (OHC) timeseries from CIGAR, ZANNA19[40], GCOS20[26], and ARANN[41]. Shades correspond to twice the ensemble standard deviation in CIGAR and GCOS. The top-left subpanel shows the yearly ocean heat content tendency (OHCT) for the 2007–2022 period, for the CIGAR and the geodetic estimate from MOHECANv5[43], GCOS20[26], CERES EBAF Ed4.2[44], and optimized CERES timeseries[28] (CERES+). The gray rectangle in the main plot indicates the period over which the OHCT is shown. **b** OHC timeseries from CIGAR and the objective analysis experiments (see the caption of Table 1 for the definition of the experiments). **c** OHC time series from CIGAR for the three latitudinal bands Southern Extra-Tropics (90°S–30°S), Tropics (30°S–30°N), and Northern Extra-Tropics (30°N–90°N).

observation-blind model simulations (as those used as background in OA-BGSIM), significantly under-estimate the inter-annual variability, especially in the Southern Ocean.

Timeseries for the three latitudinal bands (Northern and Southern Extra-Tropics, and Tropics) are shown in Fig. 1c. They indicate the largest interannual variability and trends in the extra-Tropics (particularly, in the Southern Ocean), while a steadier increase and attenuated interannual variability in the Tropics. In terms of trends, the Southern Extra-Tropics contribute more than other regions over the 1961–2022 period (0.63 W m$^{-2}$ against 0.35 and 0.37 W m$^{-2}$ for the Tropics and Northern Extra-Tropics, in units of the Earth's surface), in agreement with previous studies[48–50].

Maps of OHC's significant trends and accelerations (1961–2022) are shown in Fig. 2, together with their uncertainty. Areas of large heat accumulation (>1 W m$^{-2}$) are visible in the Southern Ocean and the Arctic, while the Tropical band shows moderate warming, larger in the Atlantic Ocean than elsewhere. Although the in-situ observational sampling is limited in polar areas and the OHC uncertainty is therein large, CIGAR indicates the high latitudes as ocean warming hotspots, in agreement with many observation- and model-based studies[51–54]. Mid-latitudes show smaller warming than elsewhere. This agrees with previously estimated 1968–2019 trend maps[55]. The uncertainty generally follows the areas of the largest trends (high latitudes and Tropics), with notable uncertainty in the North Atlantic Ocean as well. The map of ocean heat content acceleration indicates that part of the Antarctic region (Weddel Sea) and the North Atlantic Ocean (Gulf of Mexico, western boundary currents, Labrador, Greenland, and Mediterranean Seas) suffer from ocean heat content acceleration exceeding 0.4 W m$^{-2}$ dec$^{-1}$. The acceleration uncertainty is, similarly to the trend, the largest at high latitudes and near the Equator, but significantly smaller than the signal.

The large ensemble system allows us to quantify and understand recent changes and associate statistical significance with them. We show in Fig. 3 the areas that exhibit statistically significant OHC increase in 2022 compared to 2021 (last year's OHC increase), which is of great importance for climate monitoring. Large portions of the mid-latitude Southern Hemisphere, Atlantic and Pacific Oceans, and Tropics, exhibit a significant increase in OHC, with patterns in agreement with a previous study[56] (i.e., western Tropical Pacific Ocean, the southern part of the Indian Ocean, north-eastern Atlantic Ocean, etc.). The global OHC increase is driven by the accumulation of heat in many regions, rather than localized accumulation. To understand how global warming is affecting the OHC increase distribution, panel b of Fig. 3 shows (for each grid point of the large ensemble reanalysis system) the year owing the largest OHC. Large portions of all the main basins exhibit the latest two years as the warmest years, with some exceptions, for instance, located in the Eastern Tropical Pacific upwelling region, central North Atlantic Ocean, and Kuroshio extension. In terms of area-averaged values (Fig. 3c), on 11.6% of the global ocean, the year 2022 is the warmest year over the 1961–2022 record, almost doubling any previous year. These statistics confirm a robust acceleration of warming that concerns large areas of the global oceans. Other notable years (exceeding 5%) are 2021, 2016, and 2015.

## Analysis of uncertainties

The ocean heat content ensemble standard deviation (spread) from CIGAR is shown in Fig. 4a, b for the global ocean and three latitudinal bands, together with the uncertainty of the GCOS20 global ocean heat content assessment. The top panel shows the absolute uncertainty, while the middle panel shows the percent uncertainty, normalized by the OHC interannual variability (temporal standard deviation of the detrended timeseries). Tropics appear as the most uncertain latitudinal band, while the Northern Extra Tropics are the least (with uncertainty values always <30%), mostly because of the maturity of its observing network before the Argo float deployment. Peaks in the

increase (see also Table 1), evident in the poorly sampled period. When the assimilation frequency and time window are extended from ten days to one month (experiment OA-MON), variability and OHC increase are damped during the most recent, well-observed, period; an attenuation of variability during the full reanalysis period occurs when horizontal correlation length scales in the data assimilation system are halved (experiment OA-SLS). In all these experiments, interannual variability, trend, and acceleration approach those of the GCOS20 assessment. This indicates that the combined choice of the background field, correlation length-scales, and temporal frequency in the objective analysis reconstructions affect the OHC interannual variability. Maps of interannual variability (Fig. 2a, d) reveal that areas of intense mesoscale activity (e.g., the western boundary currents, and the Antarctic Circumpolar Current) are those characterized by the largest interannual fluctuations. Here, objective analyses may struggle to reproduce such fluctuations due to their coarse temporal resolution and the lack of atmospheric forcing information. Likewise,

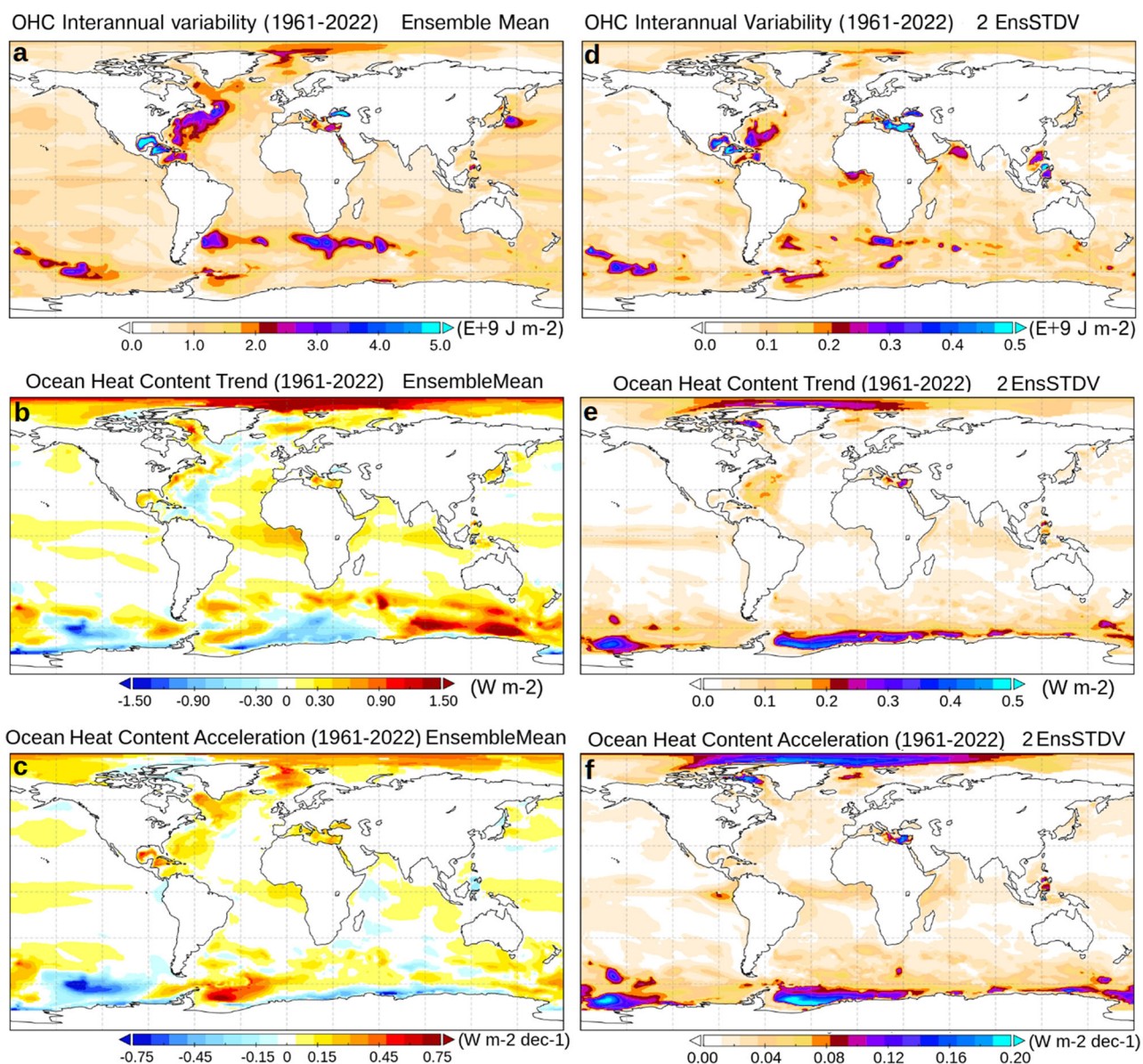

**Fig. 2 | Ocean heat content variability.** Ocean heat content interannual variability (**a**), trend (**b**), and acceleration (**c**) from the CIGAR ensemble mean, and their corresponding uncertainty (**d**–**f**), calculated as twice the ensemble standard deviation from CIGAR. The interannual variability is taken as the standard deviation calculated over the detrended yearly means.

Tropics correspond to strong El Nino events (e.g., 1998) and seem not much affected by the TAO/TRITON mooring array deployment in the 1990s. This is consistent with previous analyses[57], which show that El Niño events lead to an increase and broadening of ensemble anomalies and covariances in reanalyses compared to neutral years, due to the enhanced thermocline variability. The Southern Ocean has very large uncertainty in the 1960s, which rapidly resembles that of the global ocean (35-40%) from the 1970s onward. Overall, the global ocean shows an uncertainty of about 40%, which drops to about 15% during the last decade (2013–2022). Compared to GCOS20, our reanalysis system has often a larger uncertainty but is steady at around 40% before the 2000s, while GCOS20's fluctuates between 10% and 50%. The larger uncertainty in CIGAR is likely due to the many more sources of uncertainty in reanalyses than objective analyses (e.g., atmospheric forcing, model physics, etc.), which dominate the ensemble dispersion in periods with poor observational sampling. During the last decade, the normalized uncertainties of CIGAR and GCOS converged towards similar values of about 15%.

Temporal variations of warming uncertainty (ocean heat content trend), at the global scale, are evaluated by computing (Fig. 4c) the running ensemble standard deviation of trends, over both a 15 and a 30-year time window. The panel shows the running ensemble mean of the trend for comparison (in red, with the right-side axis). The trend uncertainty is high (more than $0.10 \, \mathrm{W \, m^{-2}}$) at the beginning of the timeseries and stabilizes between 0.05 and $0.10 \, \mathrm{W \, m^{-2}}$ until approximately 1995; then, it increases to more than $0.15 \, \mathrm{W \, m^{-2}}$ (2001) for the 15-year time interval and decreases afterward, up to values of $<0.05 \, \mathrm{W \, m^{-2}}$. The increase starts earlier than the Argo float deployment; however, it is difficult to say how much of this is affected by the observational sampling, which only partially follows the 15-year running trend variability. Additional objective analysis (OA) experiments randomly withholding 50%, 75%, and 90% of Argo floats, or with no observation assimilation below 1000 m of depth, led to statistically insignificant differences in ocean warming and acceleration compared to the OA experiment with the full observing network; this suggests that our results are not significantly influenced by the change in

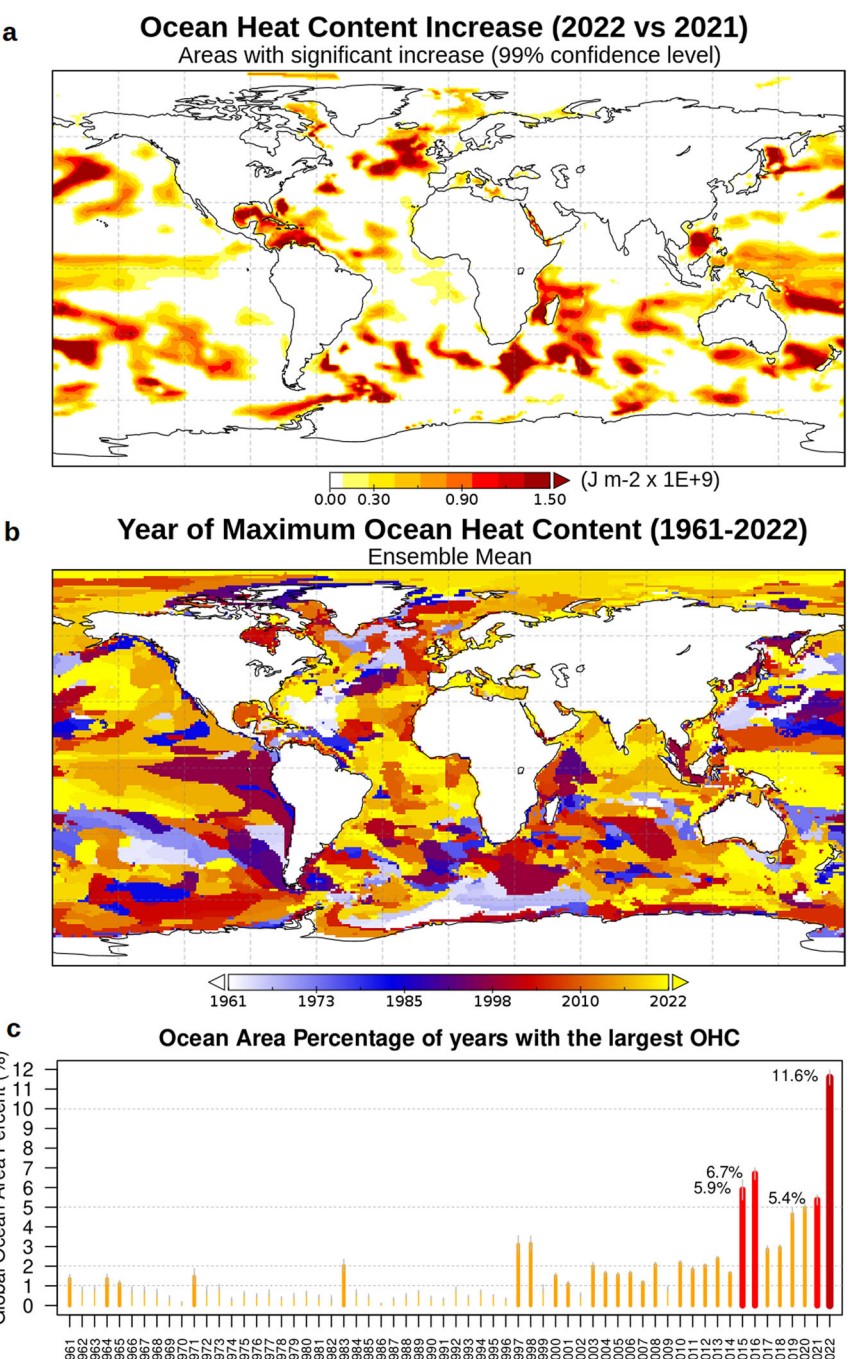

**Fig. 3 | Recent increase in ocean heat content. a** Significant increase in Ocean heat content (OHC) from 2021 to 2022 (significant cooling or non-significant differences are masked in white to emphasize the regions with heat content increase). **b** Year corresponding to the maximum yearly OHC for every grid point of CIGAR. **c** Percent of ocean surface showing maximum yearly OHC, as a function of the years from 1961 to 2022. Values are shown for the years exceeding 5%.

observational sampling following the deployment of Argo floats. Figure 4c also reports diagnostics with 30-year time intervals, which show much steadier behavior than those with 15-year intervals.

Clustering the ensemble members by grouping their type of perturbations, i.e., their sources of uncertainty, allows us to understand the main sources of uncertainties, as explored in regional operational contexts[58]. In practice, this is done by considering the averaged spread for members with the same source of uncertainty (see the "Methods" section), in comparison with the total (full ensemble) spread. We show in Fig. 5 the outcomes of this assessment in terms of both global and regional diagnostics (Fig. 5a, b), and maps of

significant prevailing sources of uncertainty (Fig. 5c, d; see the figure captions for the exact definitions).

The uncertainty of the global trend (top left panel) is a percentage estimate of the total ensemble standard deviation (which accounts for model physics and data assimilation uncertainty). In this case, the sources of uncertainty are not additive and saturate between 45% and 75%, meaning that they can all explain a large portion of the resulting ensemble dispersion. For the 1961–2022 global trend, the main source of uncertainty is the air–sea flux formulation and the observation dataset, provided that, in the global OHC trend estimates, the air–sea flux formulation directly drives the heat tendency because horizontal

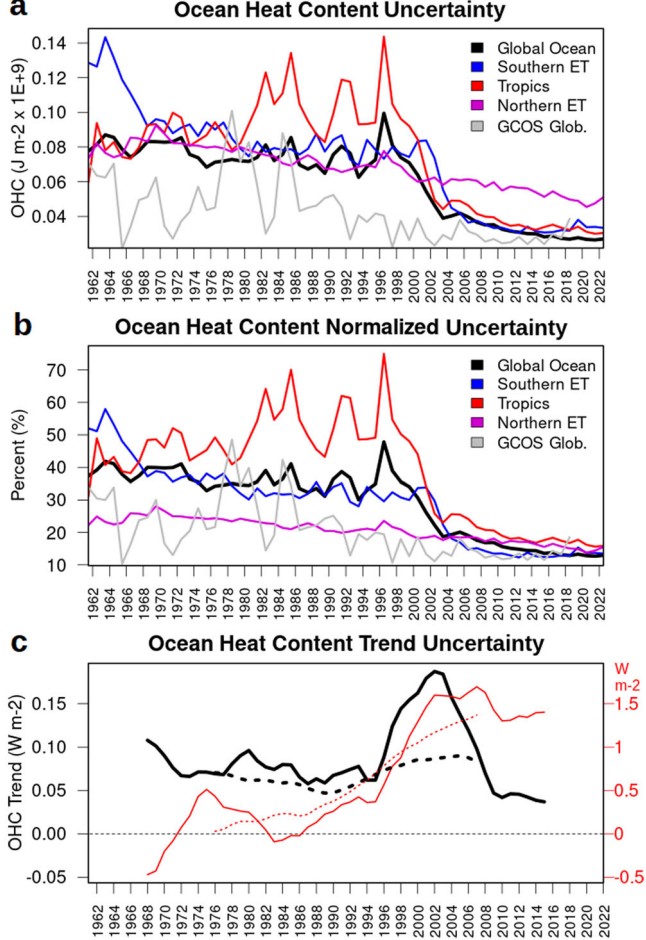

**Fig. 4 | Ocean heat content uncertainty. a, b** Yearly ocean heat content (OHC) ensemble standard deviation (spread) for the global ocean and the three latitudinal bands defined in the caption of Fig. 1. Absolute values are in panel **a**, while percent values (normalized by the interannual variability, i.e., the interannual detrended standard deviation) are shown in panel **b**. **c** Running ensemble standard deviation of OHC trends over 15-year (solid) and 30-year (dashed) time intervals. Red lines and axis report the running ensemble means of the trends.

in the North Atlantic, initial conditions uncertainty dominates in the Indian Ocean, and SST uncertainty mostly manifests at low latitudes. The atmospheric boundary conditions uncertainty confirms more marginal and ascribed to small portions of the global oceans. For the poorly sampled period 1961–2001, patterns are more complicated to interpret, with SST uncertainty extending its areas of dominance to mid-latitudes; the atmospheric boundary conditions become the prevailing source of uncertainty in large portions of the Southern Ocean.

## Discussion

With a large ensemble ocean reanalysis at moderate resolution (1/3°–1°) we have quantified global and regional ocean warming, its acceleration and confidence, evaluated its uncertainty, and identified the main sources of uncertainty. We have investigated the enhanced interannual variability and trends in the reanalysis compared to objective analyses. Sensitivity experiments show that these are not an artifact of the reanalysis system but depend closely on the choices of the assimilation frequency, background field, and error length scales, in objective analyses.

The 32-member ensemble was built with the scope of including explicitly the major sources of uncertainties (atmospheric forcing and its air-sea flux formulation, observation bias correction, initial conditions, and sea surface temperature accuracy). We combined these sources of uncertainty within the ensemble generation, adding stochastic modulation of ocean model parameters and assimilated observations in the reanalysis system. Results show that our system has similar ocean warming values and uncertainty as other dependent and independent datasets, confirming the reliability of the ensemble. Moreover, the approach allows us to understand the hierarchy of uncertainty sources in ocean heat content reconstructions. Thus, this multi-perturbation ensemble reanalysis system is promising for use over earlier historical periods, for which spanning the major sources of uncertainty in an ensemble context is crucial to correctly represent the uncertainty envelope and quantify the time-dependent signal-to-noise ratio in the ensemble system.

The 1961–2022 warming is quantified in $0.43 \pm 0.08 \, \text{W m}^{-2}$. The acceleration is found significant and equal to $0.15 \pm 0.04 \, \text{W m}^{-2} \, \text{dec}^{-1}$. Regional patterns show a dominant trend and acceleration at high latitudes and near the Equator, with mid-latitudes exhibiting less pronounced accumulation, due to the large meridional heat transports therein. Patterns of OHC increase in 2022 are well spread over all the basins, and more than 11% of the global ocean shows its highest OHC in 2022, almost doubling any previous year. Before the Argo era, the relative OHC uncertainty was the largest in the Tropical band and partly associated with El Nino events. On a global scale, the uncertainty reduced from 40% of its natural variability in the 1960s to about 15% during the last decade.

Among the different sources of uncertainty, regional trends are mostly affected by observation procedures (bias correction, especially at high latitudes) and SST data uncertainty (especially at low latitudes). In contrast, all the sources of uncertainty contribute to the global trend uncertainty. The reanalysis initialization uncertainty plays a more marginal role, except in specific regions (e.g., in the eastern Indian Ocean) where the local or remote memory of the ocean or poor observational sampling may amplify its impact.

These findings shed light on the contemporary ocean warming acceleration and foster the design of reanalyses and reconstruction systems aware of all sources of uncertainty. Ensemble reanalyses, although with a much smaller ensemble size, are increasingly used to reconstruct the climate of the past and initialize long-range predictions[59], and the uncertainty ranking provided here (at regional and global scales) will help guide the ensemble generation approach.

and vertical redistributions of heat do not affect the global mean signal. The sea surface temperature (SST) uncertainty exceeds 70% of the global trend uncertainty, while other sources show smaller values, with the initial conditions having the smallest percentage of 45%. The early period 1961–2001 shows similar results, with the atmospheric forcing uncertainty becoming the main source of uncertainty.

Regional trend uncertainty is defined as the percent area where a certain source of uncertainty is significantly prevailing. Thus, percent values are additive in this case. For the entire period, it is the largest for the observations, which account for about 22% of the uncertainty, followed by the SST uncertainty (13%) and initial conditions uncertainty (11%). Atmospheric forcing and the air–sea flux formulation together do not exceed 10% and represent more marginal sources of uncertainty than the others. In the observation poor period 1961–2001, the SST uncertainty, which plays a major role in the absence of other developed networks, is the largest source of uncertainty accounting for about 17% of the total regional uncertainty.

Maps of prevailing types of uncertainty indicate which uncertainty source is locally dominant—statistically significantly—in the trend uncertainty. Only areas where a statistically significant source occurs are shown. The uncertainty associated with the observation preprocessing (OBS) is found to prevail in the Southern Ocean and

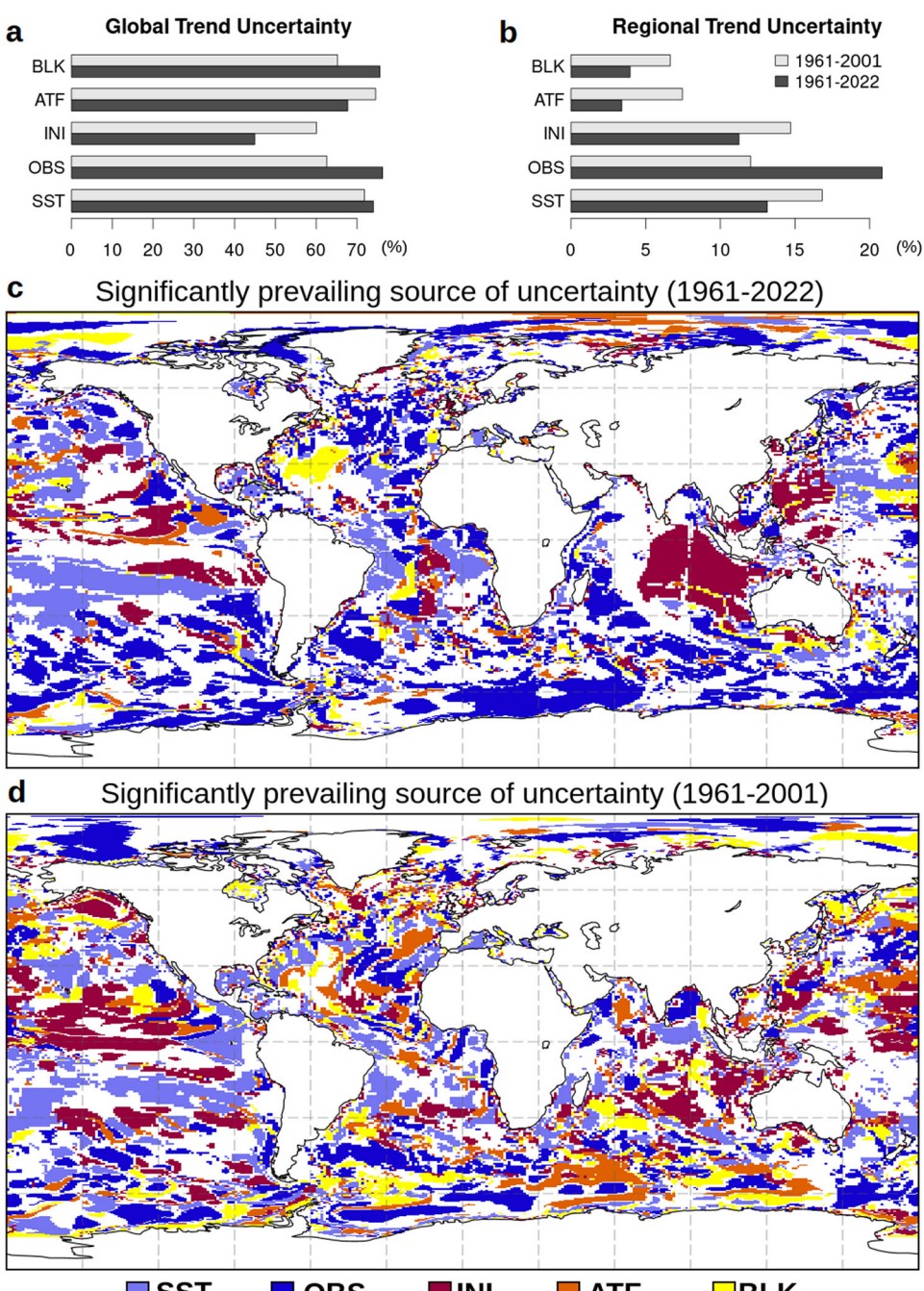

**Fig. 5 | Main sources of uncertainty in the Ocean heat content reconstruction.** **a**, **b** Percent uncertainty associated with the different sources of uncertainty for the global trend (**a**, as a percent of the individual source that explains the total uncertainty) and regional trends (**b**, as a percent of the ocean with the prevailing source of uncertainty). **c**, **d** local (statistically significant) prevailing source of uncertainty. White areas denote points with insignificant prevailing sources.

Diagnostics are shown for both the full period 1961–2022 (**c**) and the observation-poor period 1961–2001 (**d**). SST is the sea surface temperature uncertainty, OBS is the in-situ profile uncertainty, INI is the uncertainty of the initial conditions at the beginning of the reanalysis period, ATF is the atmospheric input data uncertainty, BLK is the uncertainty in the air–sea flux formulation.

## Methods
### Ocean model configuration
The ocean model is NEMO[60], version 4.0.7, which includes the five-category sea-ice dynamic and thermodynamic model SI³ [61]. It is implemented at about 1° of horizontal resolution, with augmented meridional resolution in the Tropics (up to 1/3°), and 75 vertical depth levels with partial steps[62]. The model uses bulk formulas adapted from the AERO-BULK package[63], a 3-band RGB scheme for the penetrating component of the air–sea heat flux, with extinction coefficients that depend on a monthly climatology of chlorophyll[64]; for vertical mixing, the TKE

scheme[65] is used with re-tuned background eddy viscosity and diffusivity[66], while the horizontal diffusion operator is composed by a Laplacian operator for temperature and salinity and a bi-Laplacian for momentum.

The system is forced at the surface by the ECMWF ERA5 reanalysis[67]. Hourly fields of near-surface temperature, humidity, wind, and pressure are used to calculate high-frequency turbulent fluxes, while daily means of radiation and precipitation fields are used, with an analytical diurnal modulation[68] of downwelling solar radiation. At the sea surface, heat and freshwater fluxes are corrected through a

                                                7

relaxation scheme that nudges SST and sea surface salinity (SSS) to external analyses. The relaxation time scale is set to 1 month and 1 year for SST and SSS, respectively. SSS analyses are taken from the UKMO EN4[69], while SST analyses depend on the specific ensemble member (see the section "Sources of uncertainty and ensemble generation").

The freshwater discharge onto the ocean from land and ice sheets is provided by the JMA JRA-55-do reanalysis[70], as inter-annually varying daily mean freshwater inputs, re-gridded onto the NEMO grid through conservative remapping.

## Assimilation scheme

An oceanic 3DVAR scheme is used for assimilation, which implements background-error covariances estimated from long-term climatological anomaly differences, re-tuned via a posterior calibration method[71] within preliminary assimilation experiments. In particular, background-error vertical covariances are implemented through the application of multi-variate spatially varying EOFs, while a first-order recursive filter that uses spatially varying correlation length scales is adopted horizontally[72]. The set of assimilated observations includes all in-situ profiles (MBT, XBT, and CTD casts, moorings, floats and gliders, and animal-borne sensors), extracted from the UKMO EN4 dataset[69]. Observational errors were initially taken from previous estimates[73] and calibrated employing the posterior method[71]. A variational quality control scheme[74] is used for non-linear quality control of the assimilated observations. A 10-day assimilation time window and analysis increment application frequency are adopted.

In addition to the variational assimilation scheme, a large-scale model bias correction (LSMBC) scheme is applied[75], where deep ocean waters (below 500 m of depth) are weakly relaxed towards external objective analyses, at a temporal and spatial scale of 10 years and 1000 km, respectively. Additional experiments with the withholding of the scheme, or in simulations using the LSMBC but without assimilating in-situ profiles, or comparing CIGAR with the objective analyses indicate that such a large-scale bias correction scheme has no significant impact on the resulting warming of CIGAR. Indeed, this is driven by the assimilation of in-situ profiles, in terms of long-term changes and interannual variability. The fact that the OA objective analysis experiment (see its description below), with no LSMBC, has a very close variability and trends to CIGAR, further confirms that LSMBC is negligible for ocean warming assessments.

For comparison with the reanalysis system, an objective analysis scheme (OA, hereafter) is built upon the same data assimilation scheme. The two analysis systems share the same data assimilation configuration, with an analysis frequency of 10 days, but OA does not include any dynamical ocean model (namely, the model time integration step), and simplifies to an observation statistical mapping, which uses the same variational data assimilation scheme as CIGAR. As background-error covariances are estimated from climatology anomalies, they are representative of the climate modes of variability rather than the background accuracy; thus, the same set is used for both reanalyses and objective analyses. In OA, background fields are taken from a 10-day climatology to which the 10-day (persistent) anomaly from the previous analysis cycle is added, mimicking the cumulative effect of data assimilation in reanalyses. The OA system is used to interpret the different behavior of the ocean heat content reconstructed from the ensemble reanalysis compared to that from objective analyses.

## Sources of uncertainty and ensemble generation

The large-ensemble reanalysis system CIGAR accounts for the major sources of uncertainty in the system configuration. These are listed below, and correspond to the ensemble generation strategy:

- Observation bias correction (for MBT, XBT, and floats) is an important source of uncertainty for multi-decadal retrospective analyses[15]. Here, it is spanned using two different sets of observations (namely, two different algorithms for XBT corrections) which include,

respectively, two different XBT corrections[76,77]. Both corrections are provided through the UKMO EN4 profile dataset.

- The SST dataset used in the surface relaxation scheme; here we consider two SST analysis realizations, given by the UKMO HadISST[78] and the JMA COBE[79] analyses. The differences between the two datasets exceed 1 °C in areas of enhanced mesoscale activity (e.g., around the Antarctic Circumpolar Current), and show, in terms of globally averaged values, that COBE-SST is slightly colder than HadISST by about 0.15 °C. After 2005, the two datasets tend to converge.

- Initial conditions at the beginning of the reanalyzed period are a possibly important source of uncertainty, especially for ocean heat content analyses;[31] here, we use two sets of initial conditions, extracted from lagged simulation restarts in previous assimilation experiments. Specifically, we have taken the 1948 and 1968 initial conditions from previous pilot assimilation experiments as initial conditions valid in 1958; we consider the first three years (1958–1960) as an adjustment period. The pilot assimilation experiment is the last iteration of many iterative 1958–2021 reanalyses performed as spinup to stabilize the ocean model drift, with initial conditions taken from the latest restart of the previous iteration.

- Atmospheric forcing uncertainty is accounted for using two different ensemble members of the ERA5 reanalysis. We added the ensemble anomalies from members 1 and 4 of the ERA5 ensemble data assimilation (EDA) system to the deterministic ERA5 reanalysis fields. The two members have been chosen after clustering the atmospheric surface kinetic energy following a clustering method[80].

- Air–sea flux formulation, whose uncertainty is modeled using two distinct bulk formulations, i.e., the NCEP/CORE formulas[81] and that from ECMWF[82]. The two bulk formulas implement, among others, a different formulation of the Charnock coefficient and a different use of the sea surface temperature (bulk versus skin), respectively[63,83]; moreover, within the bulk formula implementation, the NCEP uses the absolute wind while the ECMWF the relative wind (namely, the surface currents are subtracted to the wind).

The large ensemble results from any possible combination of these sources of uncertainty, resulting in a 32-member reanalysis. On top of this configuration, we use a stochastic parameter perturbation (SPP) scheme[84] to stochastically account for the model uncertainty (in particular, the uncertainty of the solar extinction coefficients, the TKE parameters, the surface nudging relaxation time, and the horizontal diffusivity and viscosity). Additionally, we perturb the observations before their ingestion in the variational data assimilation system, through a Gaussian random deviation, with the standard deviation equal to the representativeness error of the observations[85]. This allows us to consider stochastic perturbations for both the assimilation and forecast steps, in addition to the sources of uncertainty explicitly detailed above.

We estimated the sampling uncertainty of the ensemble mean of the global OHC trend (1961–2022) using previous definitions[86] and found that the 32-member ensemble size reduces by 88% the sampling error compared to a 10-member ensemble. The sampling uncertainty may further decrease with a larger ensemble than 32; however, the size of 32 is chosen as a compromise between small sampling uncertainty and computational affordability.

This study is designed to account for the major sources of uncertainty in the reanalysis system. However, there may be other processes, that may induce additional uncertainties and are not sampled by our system, for instance: exchanges of heat between the ocean and the sea ice; small-scale energy exchanges that cannot be resolved by our reanalysis system due to its limited spatial resolution; systematic errors in the atmospheric forcing not spanned by the ERA5 members; observational sampling error not spanned by the use of one observation production system (EN4).

**Improvements compared to previous reanalysis estimates and assessments**

Previous studies showed limited consistency among reanalyses in comparing the ocean heat content, with poorly reliable long-term trends. For instance, it was found that drifts and/or spurious variability below 700 m of depth degraded the accuracy of ocean warming estimates from reanalyses[35]. Such work was part of the Ocean Re-Analysis Intercomparison Project (ORA-IP[87]), which is an intercomparison exercise initiated in 2012 with a vintage of reanalyses more than 10 years old. Since then, many advances have been achieved in both modeling and data assimilation aspects of reanalyses, most of them discussed, respectively, in recent works[88,89]. Ocean model parametrizations, including extensive tuning of vertical mixing schemes and sea-ice models[66]; enhanced representation of background-error vertical correlations, improved bias-correction schemes[27]; improved estimates of air–sea fluxes[90] were all crucial upgrades of reanalysis systems leading to more reliable ocean heat content reconstructions. For instance, it was found that a successive vintage of reanalyses compared to that of ORA-IP improved the accuracy of the representation of ocean heat content variability[91], especially in extratropical regions[57].

Furthermore, the CIGAR reanalysis is specifically designed for OHC investigations, unlike many other reanalyses included in the ORA-IP intercomparison exercise[35], which were built for initializing long-term prediction systems, and not as a climate monitoring tool, and did not account for a robust spinup procedure (most of them cold-starting in 1993 without any stabilization or spinup protocol). CIGAR has been instead spun up to avoid any possible drift (see the section "Sources of uncertainty and ensemble generation"). Additionally, it is known that one of the main causes of spurious variability in reanalyses is the assimilation of altimetry[35,92], which often leads to noisy or spurious deep ocean increments when the interior ocean is not constrained by in-situ profiles (i.e., before the Argo float deployment). Our system does not include the assimilation of altimetry data because it is designed to span more than 60 years (and will be further extended in the past); consequently, it cannot reproduce any possible problem during the altimetry era.

## Data availability

OHC ensemble data from CIGAR generated in this study have been deposited in the Storto and Yang (2023) database in Zenodo [https://doi.org/10.5281/zenodo.8395297]. The latest version of the dataset includes the gridded OHC dataset together with the diagnostics on which the figures of this article are based. Description of the dataset, instructions, and contact form for requesting other variables are available through the website http://cigar.ismar.cnr.it/.

## Code availability

The ocean model code is NEMO (v4.0.7), which is available for download from https://forge.ipsl.jussieu.fr/nemo/changeset/15813/NEMO/releases/r4.0/r4.0-HEAD?old_path=%2F&format=zip. Our modifications to this version of the NEMO model (for data assimilation purposes, stochastic physics, and several other enhancements) are available from https://git.isac.cnr.it/storto/nemo_4.0.7_orca1_cnr. The variational data assimilation code is available from https://baltig.cnr.it/nemo_ismar-rm/3dvar_histrea.

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

## Acknowledgements

Acknowledgment is made for the use of ECMWF's computing and archive facilities in this research, and the extraction of the ERA5 reanalysis. The EN4 subsurface ocean temperature and salinity data were collected, quality-controlled, and distributed by the U.K. Met Office Hadley Centre. SST data were provided by the U.K. Met Office Hadley Centre and the Tokyo Climate Center of the Japan Meteorological Agency. GCOS OHC data were obtained from the DKRZ data repository.

## Author contributions

A.S. and C.Y. have designed the study, analyzed the results, and contributed to the text. A.S. ran the ensemble reanalysis system and the additional experiments.

## Competing interests

The authors declare no competing interests.
