## [Peer Review File · Nature Communications]

Acceleration of the ocean warming from 1961 to 2022
unveiled by large-ensemble reanalysesREVIEWER COMMENTS

Reviewer #1 (Remarks to the Author):

Review of Acceleration of the ocean warming from 1961 to 2022 unveiled by large-ensemble by Storto & Yang for Nature Communications

Summary and key results:

This manuscript reports accelerated ocean warming over the period 1961-2022 as evidenced by an ensemble of ocean reanalysis calculations. Resulting trends in OHC agree with previously published estimates using direct observations and objective mapping. 2022 is reported to be the warmest year of the record with 11% of the global ocean reaching a doubling compared to previous years. This work emphasizes different sources of uncertainty and their quantification.

Recommendation and significance:

This paper is an important contribution to the general field and WCRP goals (could be mentioned! <https://sites.google.com/magellium.fr/eeiassessment/objectives> or GDAP ASSESSMENTS <https://www.gewex.org/panels/gewex-data-and-analysis-panel/gdap-projects/> for more info) geared toward better understanding and quantifying uncertainty in ocean heat content and Earth energy imbalance estimates. The manuscript does neither quantitatively nor qualitatively intercompare results with other ocean state/reanalysis or satellite-based OHC estimates (more details below) – some of which indicate accelerated ocean warming, primarily since ~2015. I highly recommend to place the presented results into context with other relevant work. I therefore recommend to accept this paper after my comments below have been addressed.

Minor comments:

Line 39: Please explain or add reference to address the ‘non-uniform radiative forcing from the atmosphere’ and its implications for OHC change pattern.

Line 54-64: I recommend including satellite-based (total sea level minus ocean mass) estimates as an additional example of OHC analysis, potentially as a part of point iv)?

Line 67: I would mention the size of the ensemble early on. It came quite late in the discussion section. With respect to ensemble size 32: How have the authors determined this number? What is the optimal number to stabilize sample uncertainty and would they recommend to increase the size? I came across this paper: And wonder whether it applies here:

<https://rmets.onlinelibrary.wiley.com/doi/full/10.1002/qj.3387>

Line 89: Have you estimated acceleration in ocean warming over the more recent period? And how does it compare to recent results by, e.g., Loeb et al. and Hakuba et al.? How exactly is acceleration defined/computed?

Loeb: <https://agupubs.onlinelibrary.wiley.com/doi/10.1029/2021GL093047> report trend for the net CERES TOA energy flux is $0.50 \pm 0.47 \text{ W m}^{-2} \text{ decade}^{-1}$

Hakuba: <https://agupubs.onlinelibrary.wiley.com/doi/full/10.1029/2021GL0936240.50> report $0.25 \text{ W m}^{-2} \text{ decade}^{-1}$ over 2002-2019

Line 115 ff: Have you computed hemispheric mean OHC trends? To my knowledge, the NH does not contribute much to the observed global change, e.g., Fig. 1 here

<https://link.springer.com/article/10.1007/s40641-016-0043-9>

Line 202: Why do initial conditions uncertainty affect some regions more than others?

Line 234: With large-scale circulation are you referring to meridional heat transports, which I believe might be peaking in regions where ocean heat storage is smaller?

Line 244 and section 4.3: To my knowledge, the construction of other systems can differs from the system described here (e.g. 4Dvar). Are the results relevant to other estimation systems (e.g., ECCO, ORAS)? Would it make sense to summarize different types and what uncertainty might be relevant to those?

Line 264: I believe SSS stands for sea surface salinity, please spell out.

Reviewer #2 (Remarks to the Author):

In this manuscript the authors evaluate the ocean heat uptake (OHU) since 1961 with an ensemble of 32 reanalyses which are all based on the same ocean model and the same assimilation scheme but which use different in-situ and SST observations, different atmospheric forcing, different air-sea flux formulation and different initial conditions. The ensemble approach enables the author to evaluate the uncertainty in reanalyses' estimates of the OHU that is due to respectively observations, atmospheric forcing, air-sea flux formulation and initial conditions. The authors find an OHU of 0.43 ± 0.08 W m⁻² since 1961 which is consistent with objective analyses of in situ observations. However, they find a larger interannual variability in OHU than in objective analyses before 2001. They also find a significant acceleration in OHU of 0.15 ± 0.04 W m⁻² dec⁻¹ that is not present in objective analyses. They find that the uncertainty in reanalyses estimates' of the OHU is dominated by the uncertainty in the tropics.

The manuscript is clear and easy to follow. On the overall the manuscript is pretty well written but the it would benefit of a checked for more lucidity. The experimental design is sufficiently detailed and referenced. In addition, the manuscript deals with an important question which is the estimate, over the last decades, of the global ocean heat uptake , the regional ocean heat content and the associated uncertainty. The effort to detail and explain uncertainties is a remarkable strength of the manuscript.

The authors claim 4 important results in this study. They claim that

- 1- their ensemble of reanalyses shows the same OHU trend as objective analysis over the period 1961-2021 which, according to them, gives confidence in this estimate of the OHU
- 2- their ensemble of reanalyses shows a larger interannual variability in OHU than objective analysis before 2001 and that this signal is due to the choices of the background, the assimilation frequency, and the error length scales in the assimilation scheme
- 3- there is a significant acceleration in the reanalyses' estimate of the OHU between 1961 and 2022 which is not present in the objective analyses
- 4- the main source of uncertainty in reanalyses' estimate of the OHU comes from the tropics and regionally it is due to observational uncertainty

I have important reservations on each of these results.

Concerning result 1, I think it is quite an obvious result and it cannot be considered as a novelty or a step forward. Indeed, the reanalyses used by the authors assimilate in-situ data down to 700m depth and from 500m it relaxes (at large scale and long time scales of 10 yrs) towards an objective analysis of in situ temperature. So, it is expected that the global OHU of these reanalyses will converge towards the same multidecadal trend in OHU as the objective analysis. It is because the reanalyses used here are tied to fit the same observational data as the one used in objective analyses. On this basis, I don't think the agreement between the reanalyses' estimate and the objective analysis' estimate of the global OHU trend gives any confidence.

I find results 2, 3 and 4 much more significant than result 1 and I suggest the paper should focus on these results. However I have reservations as well on these 3 results. Concerning result 2 and 3, I noticed something strange all along the manuscript. The authors seem to take objective analyses as a reference benchmark against which they evaluate their reanalyses. This is probably sounded for the multidecadal trend in OHU that seems to be properly captured by objective analyses (see the papers from Cheng and colleagues for example) but that is certainly not sounded for interannual variability (see for example the community paper of Meyssignac et al. 2019) let alone the acceleration. If we believe the interannual variability in EEI retrieved by CERES (that is confirmed by the interannual variability in OHU derived from altimetry minus GRACE, see Hakuba et al. 2021, Marti et al. 2022) then there should be much more interannual variability since 2002 in the OHU than showed by objective analyses. So, what is the real signal in OHU interannual variability? How can we be sure the reanalyses are showing OHU interannual variability that is actually real? Can you propose any better evaluation than just comparing with objective analyses? What about comparing with CERES and Altimetry minus Grace since 2002? What about before 2001? Before 2001 the interannual variability derived from reanalyses seems very sensitive to the background. Then which background is the most appropriate and why? I think you should dig much more on these aspects.

Concerning result 4, I think your findings are representative of your reanalyses only, use a specific relaxation scheme below 500m. Other reanalyses, which do not use this scheme, show very large discrepancies (e.g. Palmer et al. 2017) due to drifts in the deep ocean. So, to which extent your results are representative of the reanalyses in general? What is the uncertainty associated to this relaxation scheme? Can we be confident in it? To which extent does this relaxation scheme ties the reanalyses to objective analyses? You should explore these aspects to give a comprehensive analysis here.

In light of these reservations, I think the paper is not ready for publication. I think the point raised above should be addressed prior publication in order to strengthen the results and the significance of the paper

Detailed comments

L8-9: "caused by human-induced climate change " you probably mean "caused by GHG". GHG emissions cause energy imbalance which further causes climate change and not the other way round.

L16: You find that reanalyses agree with objective analyses in the OHU trend. Isn't it because of the relaxation scheme below 500m which makes your reanalyses stick to objective analyses? Previous studies (e.g. Palmer et al. 2017) suggested the actual major source of uncertainty among reanalyses is the drift in unconstrained deep ocean. To which extent your reanalyses with this specific relaxation scheme

are representative of other reanalyses that do not use the same relaxation scheme?

L17: consider rephrasing in direct form

L19: "before and after" you probably mean "respectively before and after"

L20: what do you mean by "observational procedure"?

L36: ocean acidification is primarily driven by the increase of atmospheric CO₂ concentrations and not ocean heat content. Please correct

L43-44: please add a reference to justify the importance of reliable uncertainty envelopes like Meyssignac et al. 2019.

L45: remove indeed

L55: there are much more relevant references than von Schuckman et al. 2020 for this. Abraham et al. 2013, Boyer et al. 2016

L59: Resplandy et al 2019: Their paper has been corrected and the method proposed has a very large uncertainty not suitable for measuring the current EEI of only 0.8W.m⁻²

L54-60: what about the geodetic method to measure the EEI with Altimetry minus GRACE. Why don't you mention it?

L63-64: Reanalyse are also attractive because it can assimilate different sources of independent data like Argo, but also altimetry and sometimes even GRACE (like ECCO)

L68: "same data assimilation system but without the ocean general circulation model": what does that mean? This is confusing!! What do you assimilate the data to, if there is no ocean general circulation model?

L72: With your approach based on reanalyses that use a relaxation scheme in the deeper layers you ignore the uncertainty in most other reanalyses in the world that do not use this relaxation scheme. Indeed, you don't account for the uncertainty due to the drift in reanalysis deep layers that has been identified in previous studies like Palmer et al. 2017. You should explain this limitation either here or in the discussion

L88-89: this result is expected as the same data that is used in objective analysis is assimilated in the reanalysis and in regions where data is sparse the reanalysis relaxes towards an objective analysis. I think you should make it clear here that your reanalyses ensemble is actually built to reproduce the in-situ global and large scale estimate. You should explain here that is not a "result" but rather an "input". Alternatively, you can frame this as "an expected result" given the assimilation scheme and the relaxation imposed in deep layers

L89-90: I think the result on interannual variability and acceleration is really interesting. This is surprising and thus this is a "result" per se because it is unexpectedly different from objective analysis. I suggest you focus the paper on this point and explain why there is this discrepancy with more in depth analyses.

L92: Why are you introducing Zanna et al. here? You don't comment on it nor explain the differences with Zanna et al. afterwards. If you keep Zanna et al. you should comment on the differences with the reanalyses and the objective analyses. You should also consider other historical reconstructions such as Bagnell and De Vries 2022 maybe also Gebbie and Huybers 2018

L96: I find only the poorly sampled period (i.e. 1961-2001) shows a higher interannual variability in reanalyses than in objective analyses. It is worth noticing it because it could be an issue associated to the sampling of observations that is not compensated by the reanalyses scheme somehow. Anyhow, the flat interannual variability after 2001 is really puzzling. CERES suggest there is much more interannual variability in the EEI and thus in OHU than showed by Argo from 2002 to 2022. So why objective analyses are so flat after 2001? That is unclear! Why do reanalysis are as flat as objective analysis after 2001? This is also puzzling! you should dig into that. Have you compared with CERES EEI?

L107: What is meant by "the background". This is unclear for non reanalyses people. You should explain in a short sentence

L110-112: You should be more precise here. OA_mon changes the trend in the densely sampled period while OA_BGSIM dampens the variability in the poorly sampled region. It would be interesting to understand why it is so and why there is this different effect of a change in background compare to a change in assimilation frequency

L117: above all, this figures suggests the southern ocean explains most of the enhanced variability in OHU during the poorly sampled period. We would like to see the same figure with the CMIP models to see if OA_BGSIM dampen the variability before 2001 in the Southern ocean which would explain why the change in background lead to a dampened OHU interannual variability.

L125-127: Have you noted that all regions where the OHU trend and acceleration are maximum are regions where reanalyses and dynamical models in general struggle to represent the actual ocean circulation. So, can we really rely on this? Can you comment?

L132: why is it of obvious importance for climate monitoring. OHU does not indicate per-se any climate hazard! The link of OHU with marine heatwaves is indirect. Marine heat waves occur on subannual time scales and depends very much on atmosphere and ocean circulation. So I am not sure to understand the "obvious" interest of measuring regional OHU if it is not to inform models for projections and prediction. Can you explain?

L145: consider changing the word "incredible"

L160: how do you explain that the GCOS study which is using the same data as you but without a

dynamical ocean state get a smaller uncertainty. It seems odd. What is your OA uncertainty is it the same as GCOS ? What does that mean in terms of reliability of GCOS uncertainty?

L184-187: but at global scale your OHU estimate is constrained by the assimilation of in-situ data in the ocean and by the relaxation towards the objective analysis in the deep ocean no? So how is it possible that the uncertainty is the air-sea flux formulation dominates? can you comment on that?

L201: explain what is meant by “explain what is the observation correction”

L226-229: I agree but you need to explain the difference with other reanalysis and the crucial role of your relaxation scheme in deep layers

L284: this relaxation scheme is a very important part!! it explains why here for the first time reanalyse do not diverge in the deep ocean. this relaxation is potentially a large source of bias and errors. How do you estimate it? Could you make a sensitivity study so we get an idea of its effect on your OHU estimates?

Reviewer #3 (Remarks to the Author):

This manuscript describes a larger ensemble reanalysis product (ENS-ERA) built on the NEMO ocean model, which assimilates ocean and surface observations via a 3DVar data assimilation system. The focus of this study is on the reconstruction of trends and uncertainties in ocean heat content (OHC) changes over the last few decades. The ensemble generation method used in ENS-ERA, which accounts for many major uncertainties associated with OHC changes, is sound and well-considered. Based on the results from ENS-ERA reanalysis, the authors managed to quantify global and regional ocean heat changes and uncertainties. I found that the diagnostics and evaluation methods used in this study are robust and, at times, innovative. The main findings from this study are very instructive and are consistent with other similar studies such as GCOS estimations.

There are, however, a few remarks that I would like the authors to address

After all, it is not possible to account for all sources of uncertainty associated with OHC changes in any model based reanalysis system. Many processes (physical/biogeochemical/volcanic), are not represented in the numerical model used by the ENS_ERA system in this study. As an uncoupled ocean reanalysis, uncertainties related to atmospheric-ocean coupling are not accounted for. Additionally, there is no mention of uncertainties associated with sea-ice modeling and heat exchanges between ocean and sea-ice models.

The ensemble generation method used in this study is sound and accounts for many major uncertainties,

including atmospheric forcings, observations, the NEMO ocean model, and initial conditions. However, limitations in the reconstruction of ocean heat content changes remain due to systematic biases in the ERA5 forcing fluxes (e.g. Cucchi, 2020: WFDE5: bias-adjusted ERA5 reanalysis data for impact studies), common characteristics in the observation datasets (e.g., the same production system except for the bias correction used in the EN4 dataset), and/or perturbation methods (e.g., how to select forcing/obs/SST/bulk formula in each ensemble member). These shortcomings should have been acknowledged in the paper.

One of the major uncertainties associated with both ocean reanalysis and objective analysis when considering OHC changes, is the sampling errors in the heterogeneous ocean observing network, which has evolved dramatically over the last few decades. E.g., the Southern Ocean is only properly sampled after Argo deployment since mid-2000. It is possible that the acceleration of a warming trend in global OHC after the 2000s is a result of better sampling (both horizontally and vertically) of the global ocean. This signal is visible in Fig.1, but was not fully discussed in this manuscript.

Detailed remarks

Abstract

L8: Here, energy imbalance refers to the ocean's energy imbalance resulting from energy input/output through its boundaries (i.e., atmosphere, land, sea-ice, and ice shelf). It is not necessarily a result of human-induced climate change, as stated in this sentence.

Instead, we can point out that ocean heat content changes is a critical climate change indicator.

L13: "all sources of uncertainties". This is a very strong statement, see my main remarks.

L20-21: I suggest to clarify "observation procedure" and "surface data uncertainty" here.

Ch. 1 Introduction

L33-37: I think this is a bit of an overstatement. It gives the impression that all changes in ocean heat content are solely due to anthropogenic causes, specifically the increase of greenhouse gases in the atmosphere induced by humans. However, other natural causes are associated with ocean heat content changes as well, namely volcanic and El Nino events.

L39: due to the uniform radiative forcing "and surface wind variability"

L44-48: Please clarify here what product are you referring to, objective analysis, ocean reanalysis, or real physical ocean?

L49-52: Uncertainties in reanalysis-based reconstructions can also arise from systematic model biases, not just vertical physics. I also suggest replacing "data assimilation configurations" with "data assimilation systems".

L54-57: It's better to clarify that ocean reanalysis is not just the use of an OGCM to simulate the ocean state. In addition, it involves using observations to constrain the projected ocean state through various data assimilation methods.

L67: Please clarify what you mean by "all major sources of uncertainty" ?

L68: For "objective analysis counterpart" add ref to Ch 4.2

Ch. 2 Results

L81-82: please confirm that the global mean ocean warming rate of $0.43 \pm 0.08 \text{ W m}^{-2}$ is averaged over

all earth surface (ocean/land/lakes) or just over sea surface ? “Earth’s area” implies this value is averaged over all surface.

L91: should be “anomalies” of global ocean heat content ...

L95-99: Do you know why the ENS-REA reanalysis show enhanced interannual variability with a sharper increase of positive trend after 2000? Is this something to do with Argo deployment since 2000s, which led to better sampling of the global ocean?

L106-107: To me, the similarity between OHC trend/interannual variability between OA and ENS-ERA suggests that the OHC signal is dominated by the input observation data sets (EN4) and is less sensitive to the model background in your system.

L109-112: The Fig.1-middle panel shows that OA-MON and OA-SLS is more close to ENS-ERA (except after 2012), with the same sharper increases of positive trend after 2000. Only after 2012 OA-MON and OA-SLS signals are closer to CGOS20.

L117: Any reason why larger interannual variability/trends in the Southern Ocean?

L120: Here “uncertainty” refers to 1 ensemble standard deviation. Uncertainty in Fig.1, however, refers to twice the ensemble std.

L119: Have you compared ocean heat content trend maps from ENS-REA (Fig.2) to that derived from CGOS20 data set? or other OA products ? Would be great if you can add a few sentences here.

L160-161: please clarify “our reanalysis system ... but is much steadier and more robust over time”.

L593: running “mean” ensemble std ...?

Ch. 3 Discussion

L210: Not sure about the “unprecedented”. Suggest to rephrase.

L213-214: See my main remakes. Ensemble generation method used in this study captures many but not all uncertainty sources (e.g. there is no sampling error in ocean observing network). I suggest to rephrase this sentence.

L218: Again, not sure about “for the first time”. Also suggest to replace “the main sources” by “many major sources”.

L227: please rephrase regarding to “all sources of uncertainty”.

L237: Could you elaborate a bit on relationship between high uncertainties in the Tropics and El Nino events?

Ch. 4 Methods

L311-314: please clarify on two sets of initial conditions used in the ENS-REA system.

Response to the Reviewers' comments.
Manuscript NCOMMS-23-08971 by Storto and Yang

Dear Reviewers,

First, we would like to thank you for the careful reading, the encouraging comments, and the many suggestions to improve the robustness and the readability of the manuscript. Below, we provide a point-by-point answer to your comments (reviewers in bold, our answer in normal font). Both authors agree with the changes. Figures in this Response are numbered as R1, R2, etc., while line numbering refers to the revised version (unless explicitly stated).

Reviewer #1 (Remarks to the Author):

Summary and key results:

[...] **The manuscript does neither quantitatively nor qualitatively intercompare results with other ocean state/reanalysis or satellite-based OHC estimates (more details below) – some of which indicate accelerated ocean warming, primarily since ~2015. I highly recommend to place the presented results into context with other relevant work. I therefore recommend to accept this paper after my comments below have been addressed.**

Thanks for the encouraging comments. We have addressed your main comments, adding the calculation for the shorter period as well, and adding missing points/methods. We have also added more datasets for comparison in Figure 1/Table 1, to make the work better placed in the context of several other OHC assessments. Please see below for details.

Minor comments:

Line 39: Please explain or add reference to address the ‘non-uniform radiative forcing from the atmosphere’ and its implications for OHC change pattern.

Thanks, it was somehow ambiguous, and we rephrased the sentence, citing Fasullo & Trenberth, 2008. “The ocean is characterized by non-uniform warming (e.g., Cheng et al., 2022), due, in turn, to the spatially and temporally varying heat uptake from the atmosphere and surface wind variability (e.g., Fasullo and Trenberth, 2008), combined with multi-scale redistribution through advective processes and vertical mixing.”

Line 54-64: I recommend including satellite-based (total sea level minus ocean mass) estimates as an additional example of OHC analysis, potentially as a part of point iv)?

We added the geodetic approach, also in Fig. 1; thanks for pointing out this previously missing, but very important, approach.

Line 67: I would mention the size of the ensemble early on. It came quite late in the discussion section.

With respect to ensemble size 32: How have the authors determined this number? What is the optimal number to stabilize sample uncertainty and would they recommend to increase the size? I came across this paper: And wonder whether it applies here:
<https://rmets.onlinelibrary.wiley.com/doi/full/10.1002/qj.3387>

Thanks for suggesting investigating this issue in more detail. The number 32 was given by the construction of the ensemble (combining different perturbations) to have a reasonably large but still computationally affordable size. We have now investigated the sampling uncertainty of the ensemble mean as defined by the paper of Leutbecher that you suggested, applied to the global OHC trend, from a randomization procedure applied to several ensemble sizes from 10 to 32. (See Fig R1 below). The sampling uncertainty decreases fast. The ~32 member shows a decrease of 94% and 88% compared to the 4 and 10 ensemble size case, respectively (which represents state-of-the-art ensemble size such as ECMWF/ORA or CERA, or SODA). We added a sentence on this in section 4.

Line 89: Have you estimated acceleration in ocean warming over the more recent period? And how does it compare to recent results by, e.g., Loeb et al. and Hakuba et al.? How exactly is acceleration defined/computed? Loeb: <https://agupubs.onlinelibrary.wiley.com/doi/10.1029/2021GL093047> report trend for the net CERES TOA energy flux is $0.50 \pm 0.47 \text{ W m}^{-2} \text{ decade}^{-1}$ (mid-2005 to mid-2019)

Hakuba: <https://agupubs.onlinelibrary.wiley.com/doi/full/10.1029/2021GL0936240.50> report $0.25 \text{ W m}^{-2} \text{ decade}^{-1}$ over 2002-2019

Thanks for suggesting these references: we added a comparison for the shorter period (lines 103-107), which highlights agreement within error bars, with our reanalysis system showing smaller uncertainty. We have also shown a comparison in terms of OHC tendencies with recent datasets (Fig 1 top panel, lines 116-122).

Line 115 Have you computed hemispheric mean OHC trends? To my knowledge, the NH does not contribute much to the observed global change, e.g., Fig. 1 here <https://link.springer.com/article/10.1007/s40641-016-0043-9>

Thanks, we added the computation (and the reference) which confirms the previous literature (lines 150-157).

Line 202: Why do initial conditions uncertainty affect some regions more than others?

In general, where the local or remote "memory of the ocean" plays an important role, or where the observational sampling is particularly poor. Specifically, it is difficult to answer which is the precise mechanism for which initial conditions play an important role e.g. in the eastern Indian Ocean. Being that these arguments are just our speculation, we add only a generic sentence on that in the Discussion lines (288-291).

Line 234: With large-scale circulation are you referring to meridional heat transports, which I believe might be peaking in regions where ocean heat storage is smaller?

Yes exactly, we rephrased the sentence accordingly (line 280).

Line 244 and section 4.3: To my knowledge, the construction of other systems can differs from the system described here (e.g. 4Dvar). Are the results relevant to other estimation systems (e.g., ECCO, ORAS)? Would it make sense to summarize different types and what uncertainty might be relevant to those?

Thanks for this point. We added a sentence on that (last paragraph of Discussion); indeed, many reanalyses (SODA, ORAS, etc.) are small-sized ensemble reanalyses, and our study may help to design the ensemble generation. The findings do not probably directly affect how ECCO (i.e., a deterministic long window 4DVAR reanalysis) can be improved, but indirectly which are the main sources of errors one should take care of; we did not include this last point which seems too specific to us, although very interesting.

Line 264: I believe SSS stands for sea surface salinity, please spell out.

Done, also for SST, at first occurrences.

Figure R1. Sampling uncertainty of the ensemble mean of the global OHC trend (1961-2022), calculated for a different ensemble size from 4 to 32. The definition is as in Leutbecher (2019), and we used a randomization procedure that relies on 1000 different ensemble member combinations for each ensemble size.

Reviewer #2 (Remarks to the Author):

Thanks for the encouraging comments and the many suggestions to improve the manuscript and the presentation of the results. Below, we address each point.

The authors claim 4 important results in this study. They claim that

1- their ensemble of reanalyses shows the same OHU trend as objective analysis over the period 1961-2021 which, according to them, gives confidence in this estimate of the OHU

2- their ensemble of reanalyses shows a larger interannual variability in OHU than objective analysis before 2001 and that this signal is due to the choices of the background, the assimilation frequency, and the error length scales in the assimilation scheme

3- there is a significant acceleration in the reanalyses' estimate of the OHU between 1961 and 2022 which is not present in the objective analyses

4- the main source of uncertainty in reanalyses' estimate of the OHU comes from the tropics and regionally it is due to observational uncertainty

I have important reservations on each of these results.

Concerning result 1, I think it is quite an obvious result and it cannot be considered as a novelty or a step forward. Indeed, the reanalyses used by the authors assimilate in-situ data down to 700m depth and from 500m it relaxes (at large scale and long time scales of 10 yrs) towards an objective analysis of in situ temperature. So, it is expected that the global OHU of these reanalyses will converge towards the same multidecadal trend in OHU as the objective analysis. It is because the reanalyses used here are tied to fit the same observational data as the one used in objective analyses. On this basis, I don't think the agreement between the reanalyses' estimate and the objective analysis' estimate of the global OHU trend gives any confidence.

Regarding result 1, we do agree is not necessarily surprising, but it is still worth mentioning, as many high-level reports (e.g. IPCC) still under-exploit the reanalyses. In any case, we have softened this point throughout the manuscript, mentioning this result as "expected" (e.g. at line 102).

The reviewer is pointing out many times in his/her review to the potentially important impact of the deep ocean relaxation (ie the large-scale bias correction scheme). However, its impact is marginal for several reasons and according to many tests. We report below several arguments (we did not think it is an important point in the original version of the manuscript):

- i) the relaxation is at a large scale and below about 700m (with a smoothed transition, see Figure R2, from about an infinite timescale at the surface to 50 years at 500m, to 10 years at 1000m), meaning that mostly relaxes very weakly to EN4 below 1000m, and the upper ocean, where observations exist, is free. Therefore the relaxation acts only on the unconstrained deep ocean, at very large scales;
- ii) the **OA** experiment is not relaxed, as it does not include any model integration. As **OA** has trend and interannual variability very close to **ENS-REA**, this is robust proof why the relaxation is not impacting the reanalysis, but the in-situ profile observations shape the OHC variability;
- iii) compared to EN4 (objective analyses), the signal of the warming is significantly different (28% weaker warming and 38% weaker acceleration in EN4 compared to ENS-REA), and shows less interannual variability, which again proves that the relaxation has a negligible role in the warming estimates, and the two datasets lead to fairly independent OHC realizations (see the Figure R3 and Table R1);
- iv) we did not detail the spinup procedure (we thought it is too technical), however, we add its description now in the revised version: we have run the reanalysis system several cycles for the period 1958-2000 until convergence (for one member only), before starting the final reanalysis production. This ensures that no drift is present in the reanalysis system. The Palmer et al. (2017) paper represents a more than 10-year-old vintage of reanalyses; modeling and data assimilation have advanced throughout this decade. Furthermore, our reanalysis (spinup, data assimilation, etc.) is specifically designed for OHC investigations, unlike many reanalyses included in the Palmer et al. (2017) paper, which were built only for initializing long-term prediction systems and not as a climate monitoring tool, and did not account specifically for a good spinup procedure. Finally, it is known (see Palmer et al., 2017; Storto et al., 2017 and many more) that the main cause of spurious variability in reanalyses is due to the assimilation of altimetry, which often leads to spurious vertical increments when not constrained by in-situ profiles. As our system does not assimilate altimetry because it is designed to span more than 60 years (and will be further extended in the past), this issue with altimetry cannot show up. Also, note that any modern reanalysis system (ORA, FOAM, CGLORS) has a bias-correction component, whose formulation may vary.

- v) we performed two important additional experiments (reported in Table R1 and Figure R3) to show that the impact of the relaxation is marginal: one control simulation (no insitu assimilation) with the large-scale bias correction, and one member reanalysis without the large-scale bias correction. One can easily verify that the impact of withholding the relaxation is insignificant (REAM1 vs REAM1-NOLSBC), while the impact of the variational assimilation of in-situ profiles (SIM-LSBC vs REAM1) significantly shapes the variability and long-term changes in ENS-REA. The relaxation is not affecting the resulting variability and related diagnostics, within the error bars (given in Table of the manuscript).
- vi) We have implemented the relaxation as a conservative procedure to avoid XBT data at depth providing spurious salinity corrections since the variational data assimilation system is multivariate and cannot warrant that salinity increments induced by temperature-only profiles are always correct.

We have added some discussion about relaxation and its impact in **Section 4** (section 4.2 at lines 338-345). We believe that all this discussion about the deep-ocean relaxation/large-scale bias correction is rather technical and does not need to be included in the main text, but we provide anyway more details in Section 4, in the revised version of the manuscript, following the concepts reported here in this reply.

I find results 2, 3 and 4 much more significant than result 1 and I suggest the paper should focus on these results. However I have reservations as well on these 3 results. Concerning result 2 and 3, I noticed something strange all along the manuscript. The authors seem to take objective analyses as a reference benchmark against which they evaluate their reanalyses. This is probably sounded for the multidecadal trend in OHU that seems to be properly captured by objective analyses (see the papers from Cheng and colleagues for example) but that is certainly not sounded for interannual variability (see for example the community paper of Meyssignac et al. 2019) let alone the acceleration. If we believe the interannual variability in EEI retrieved by CERES (that is confirmed by the interannual variability in OHU derived from altimetry minus GRACE, see Hakuba et al. 2021, Marti et al. 2022) then there should be much more interannual variability since 2002 in the OHU than showed by objective analyses. So, what is the real signal in OHU interannual variability? How can we be sure the reanalyses are showing OHU interannual variability that is actually real? Can you propose any better evaluation than just comparing with objective analyses? What about comparing with CERES and Altimetry minus Grace since 2002? What about before 2001? Before 2001 the interannual variability derived from reanalyses seems very sensitive to the background. Then which background is the most appropriate and why? I think you should dig much more on these aspects. Concerning result 4, I think your findings are representative of your reanalyses only, use a specific relaxation scheme below 500m. Other reanalyses, which do not use this scheme, show very large discrepancies (e.g. Palmer et al. 2017) due to drifts in the deep ocean. So, to which extent your results are representative of the reanalyses in general? What is the uncertainty associated to this relaxation scheme? Can we be confident in it? To which extent does this relaxation scheme ties the reanalyses to objective analyses? You should explore these aspects to give a comprehensive analysis here. In light of these reservations, I think the paper is not ready for publication. I think the point raised above should be addressed prior publication in order to strengthen the results and the significance of the paper

Please see above for the discussion on relaxation. We have now revised the manuscript discussing the recent warming and comparing it with independent estimates (geodetic approach, CERES data). This comparison provides similar behavior, and it is added in Figure 1 (subpanel zoom, discussed at lines 116-122).

The discussion on the background is improved: the middle panel of Figure 2 indicates that the choice of the background affects the OA configuration; however, in a reanalysis, the background is defined as previous model integration and cannot be changed (it would not be a reanalysis then). So, the concept of sensitivity to the background applies only to OAN and not to REAN, by construction. We have now revised the manuscript (section 2.1) to better highlight this concept and make it clear that the sensitivity studies are designed to justify that objective analyses have smoothed interannual variability compared to reanalysis. Also, in light of the new panels of Figure 2, mesoscale areas are mostly responsible for the enhanced interannual variations. We believe that the temporal (and spatial) resolution of most objective analysis systems (and the lack of atmospheric forcing) are not suitable to capture the variability in mesoscale active regions.

Finally, adding more datasets for comparison shows that our reanalysis agrees well with all these other datasets, and still holds an important interannual variability.

Experiment	Trend 1961-2022	Interannual variability	Acceleration 1961-2022
SIMUL-LSBC	0.48	0.06	0.04
EN4	0.29	0.14	0.10
REAM1	0.40	0.22	0.16
REAM1-NOLSBC	0.39	0.27	0.19

Table R1. Diagnostics (as in Table 1 of the main revised text) for the experiment: SIMUL-LSBC (a model simulation, ie no data assimilation, but with the large-scale bias correction switched on); 2) EN4 objective analyses; 3) REAM1: one-member of ENS-REA; 4) REAM1-NOLSBC: as REAM1 but without the large-scale bias-correction. Note: these are not ensemble so the confidence level (twice the ensemble standard deviation) cannot be estimated as in the main text.

Figure R2. Relaxation coefficient and corresponding timescale in open ocean as a function of depth (m).

Figure R3. Global ocean heat content anomaly 1961-2022 for the experiments explained in Table R1.

Detailed comments

L8-9: “caused by human-induced climate change “ you probably mean “caused by GHG”. GHG emissions cause energy imbalance which further causes climate change and not the other way round.

Yes thanks, we modified the text by also implementing the suggestions from Reviewer 3. (“caused by anthropogenic climate-altering gas emissions”).

L16: You find that reanalyses agree with objective analyses in the OHU trend. Isn't it because of the relaxation scheme below 500m which makes your reanalyses stick to objective analyses? Previous studies (e.g. Palmer et al. 2017) suggested the actual major source of uncertainty among reanalyses is the drift in unconstrained deep ocean. To which extent your reanalyses with this specific relaxation scheme are representative of other reanalyses that do not use the same relaxation scheme?

Please see the answer above. I would add that most reanalyses in the Palmer et al. (2017) paper were not designed for long-term OHC changes but in most cases to initialize long-term prediction systems. They started directly in 1993 without any spinup, stabilization procedure, or bias correction. Furthermore, more than 10 years of reanalysis developments and improvements should be considered as well (the vintage of reanalyses in Palmer's paper was 2012 in most cases, or even older in some cases). This was highlighted also in a recent workshop at ESA (<https://eo4society.esa.int/event/earth-energy-imbalance-assessment-workshop-2023/>) where we, as reanalysis producers, clearly demonstrated that current reanalysis products do not exhibit the problems found in the Palmer et al. (2017) paper.

L17: consider rephrasing in direct form

Sorry, but I can't see any passive form at line 17. Maybe the one at line 20. Since we are not sure and it is probably not so important we leave the abstract unchanged in these regards (in case please let us know your suggestion in the follow-up review).

L19: “before and after” you probably mean “respectively before and after”

Corrected

L20: what do you mean by "observational procedure"?

Corrected, see also answer to Reviewer 3.

L36: ocean acidification is primarily driven by the increase of atmospheric CO2 concentrations and not ocean heat content. Please correct

Thanks for spotting this inconsistent sentence, we reformulated the sentence (lines 34-38).

L43-44: please add a reference to justify the importance of reliable uncertainty envelopes like Meyssignac et al. 2019.

Added, thanks.

L45: remove indeed

Corrected

L55: there are much more relevant references than von Schuckman et al. 2020 for this. Abraham et al. 2013, Boyer et al. 2016

Added, thanks for the suggested references.

L59: Resplandy et al 2019: Their paper has been corrected and the method proposed has a very large uncertainty not suitable for measuring the current EEI of only 0.8W.m-2

The most recent paper (namely, not the retracted one) reports a warming of 0.80 ± 0.49 W in the period 1991-2016, which is still statistically significant. In any case, we refer to the paper only as an example of indirect observations leading to ocean heat content estimates, without, at this stage of the manuscript, any consideration of uncertainty. Thus, we prefer to leave the method referenced as it is now.

L54-60: what about the geodetic method to measure the EEI with Altimetry minus GRACE. Why don't you mention it?

Yes, we added it now (line 68, as also suggested by Reviewer 1). It was an important missing method, indeed. We also show it in the new subpanel of Fig. 1.

L63-64: Reanalyse are also attractive because it can assimilate different sources of independent data like Argo, but also altimetry and sometimes even GRACE (like ECCO)

Thanks, added

L68: “same data assimilation system but without the ocean general circulation model”: what does that mean? This is confusing!! What do you assimilate the data to, if there is no ocean general circulation model?

This is reformulated: it is a system with no ocean model as in objective analysis. The same data assimilation system is used but with a background coming from climatology plus persistent anomaly. Please see section 4 for details.

L72: With your approach based on reanalyses that use a relaxation scheme in the deeper layers you ignore the uncertainty in most other reanalyses in the world that do not use this relaxation scheme. Indeed, you don't account for the uncertainty due to the drift in reanalysis deep layers that has been identified in previous studies like Palmer et al. 2017. You should explain this limitation either here or in the discussion

Please see the answer above and the results shown in Figure R3.

L88-89: this result is expected as the same data that is used in objective analysis is assimilated in the reanalysis and in regions where data is sparse the reanalysis relaxes towards an objective analysis. I think you should make it clear here that your reanalyses ensemble is actually built to reproduce the in-situ global and large scale estimate. You should explain here that is not a "result" but rather an "input". Alternatively, you can frame this as "an expected result" given the assimilation scheme and the relaxation imposed in deep layers

Please see the discussion above: we do not think this is necessary "input", provided the small impact of the relaxation procedure, and only at depth. However, we do agree that such a result should be expected, at least because the systems ingest the same set of observations. We now mention this as an expected result (e.g. at line 102).

L89-90: I think the result on interannual variability and acceleration is really interesting. This is surprising and thus this is a "result" per se because it is unexpectedly different from objective analysis. I suggest you focus the paper on this point and explain why there is this discrepancy with more in depth analyses.

Thanks for the suggestion, we have now added a more detailed discussion and a map (to complement Figure 2) on the interannual variability, and an associated discussion in the text (section 2.1). Areas with strong mesoscale activity dominate the global signature, implying that products such as objective analyses (with a monthly frequency of calculation and ignoring atmospheric forcing) are by construction not adequate to represent the variability in these areas.

L92: Why are you introducing Zanna et al. here? You don't comment on it nor explain the differences with Zanna et al. afterwards. If you keep Zanna et al. you should comment on the differences with the reanalyses and the objective analyses. You should also consider other historical reconstructions such as Bagnell and De Vries 2022 maybe also Gebbie and Huybers 2018

Thanks, we added the Bagnell and De Vries ("ARANN") OHC estimate and discuss their and Zanna's results. Also, Table 1 reports basic statistics from these timeseries as well. Please note, we prefer not to include Gebbie and Huybers timeseries, as their dataset has a 5-year temporal resolution and cannot be compared to the others in terms of e.g. interannual variability, etc.

L96: I find only the poorly sampled period (i.e. 1961-2001) shows a higher interannual variability in reanalyses than in objective analyses. It is worth noticing it because it could be an issue associated to the sampling of observations that is not compensated by the reanalyses scheme somehow. Anyhow, the flat interannual variability after 2001 is really puzzling. CERES suggest there is much more interannual variability in the EEI and thus in OHU than showed by Argo from 2002 to 2022. So why objective analyses are so flat after 2001? That is unclear! Why do reanalysis are as flat as objective analysis after 2001? This is also puzzling! you should dig into that. Have you compared with CERES EEI?

We added the comparison (Fig 1 top panel, lines 116-122), which shows that for recent years the interannual variability is not flat and comparable to other datasets. Over the full timeseries, the recent interannual variability is hidden, graphically, by the enhanced warming.

L107: What is meant by "the background". This is unclear for non reanalyses people. You should explain in a short sentence

We added an explanation, "namely, the prior estimate used in the statistical analysis "

L110-112: You should be more precise here. OA_mon changes the trend in the densely sampled period while OA_BGSIM dampens the variability in the poorly sampled region. It would be interesting to understand why it is so and why there is this different effect of a change in background compare to a change in assimilation frequency

Thanks, we have reformulated the paragraph accordingly.

L117: above all, this figures suggests the southern ocean explains most of the enhanced variability in OHU during the poorly sampled period. We would like to see the same figure with the CMIP models to see if OA_BGSIM dampen the variability before 2001 in the Southern ocean which would explain why the change in background lead to a dampened OHU interannual variability.

Thanks for the suggestion, Figure R4 shows the diagnostics. We see that the damping of variability occurs globally, particularly (but not only) in the Southern Ocean as the reviewer suggested. Still, mesoscale areas in the Northern Hemisphere are visible, but much less attenuated, and very small variability in the ACC. We don't think it is worth adding this panel, but we commented on this issue in section 2.1 (lines 145-149).

Figure R4. Interannual variability map in ENS-REA (left, as top left panel of the new Figure 2), and from the CMIP-based background in OA-BGSIM (right).

L125-127: Have you noted that all regions where the OHU trend and acceleration are maximum are regions where reanalyses and dynamical models in general struggle to represent the actual ocean circulation. So, can we really rely on this? Can you comment?

Not necessarily struggling (at least for the Southern Ocean, for instance). To reply also to Reviewer 3 about this, we added the reference to the 1968-2019 trend maps of Johnson and Lyman (2020), which (at least qualitatively) show the same for Southern Ocean regions, and we refer to several publications indicating the high latitude as ocean warming hotspots (lines 161-163). The Arctic warming acceleration is due to the concurrent atmospheric warming and the increase of northward heat transports from the North Atlantic Ocean (their relative contributions being debated in the climate community); both processes can be captured by any reanalysis system.

L132: why is it of obvious importance for climate monitoring. OHU does not indicate per-se any climate hazard! The link of OHU with marine heatwaves is indirect. Marine heat waves occur on subannual time scales and depends very much on atmosphere and ocean circulation. So I am not sure to understand the "obvious" interest of measuring regional OHU if it is not to inform models for projections and prediction. Can you explain?

OHC per se is directly linked to the TOA EEI, and thus it is an important climate indicator (see e.g. <https://www.nature.com/articles/nclimate2876>). We did not refer to any MHW discussion here. We changed "obvious" to "great" (lines 174,175): we agree that "obvious" is however too strong.

L145: consider changing the word "incredible"

Changed to "robust"

L160: how do you explain that the GCOS study which is using the same data as you but without a dynamical ocean state get a smaller uncertainty. It seems odd. What is your OA uncertainty is it the same as GCOS? What does that mean in terms of reliability of GCOS uncertainty?

We believe that this is due, in early periods where in-situ data cannot fully constrain the OHC reconstruction, to the fact that the reanalysis has many more degrees of freedom and thus, uncertainties (SST data, atmospheric forcing, model physics). We don't think the uncertainty estimates in GCOS are not correct (although the temporal fluctuations are suspicious), it is just they refer to different methodologies. We have added a sentence about this.

L184-187: but at global scale your OHU estimate is constrained by the assimilation of in-situ data in the ocean and by the relaxation towards the objective analysis in the deep ocean no? So how is it possible that the uncertainty is the air-sea flux formulation dominates? can you comment on that?

See above the discussion on the marginal impact of the relaxation. Here we just stress that for global estimates, it is reasonable that air-sea flux uncertainty is very important since horizontal advection and vertical redistribution of heat do not affect the global signal. We have explicitly added this concept in the text (lines 206-209).

L201: explain what is meant by "explain what is the observation correction"

Thanks, it was misleading wording indeed. We changed it to “The uncertainty associated with the observation preprocessing (OBS)”

L226-229: I agree but you need to explain the difference with other reanalysis and the crucial role of your relaxation scheme in deep layers

Please see the answer above about the impact of the relaxation scheme.

L284: this relaxation scheme is a very important part!! it explains why here for the first time reanalyse do no diverge in the deep ocean. this relaxation is potentially a large source of bias and errors. How do you estimate it? Could you make a sensitivity study of its effect on your OHU estimates?

Please see the answer above about the impact of the relaxation scheme.

Reviewer #3 (Remarks to the Author):

There are, however, a few remarks that I would like the authors to address

After all, it is not possible to account for all sources of uncertainty associated with OHC changes in any model based reanalysis system. Many processes (physical/biogeochemical/volcanic), are not represented in the numerical model used by the ENS_ERA system in this study. As an uncoupled ocean reanalysis, uncertainties related to atmospheric-ocean coupling are not accounted for. Additionally, there is no mention of uncertainties associated with sea-ice modeling and heat exchanges between ocean and sea-ice models.

The ensemble generation method used in this study is sound and accounts for many major uncertainties, including atmospheric forcings, observations, the NEMO ocean model, and initial conditions. However, limitations in the reconstruction of ocean heat content changes remain due to systematic biases in the ERA5 forcing fluxes (e.g. Cucchi, 2020: WFDE5: bias-adjusted ERA5 reanalysis data for impact studies), common characteristics in the observation datasets (e.g., the same production system except for the bias correction used in the EN4 dataset), and/or perturbation methods (e.g., how to select forcing/obs/SST/bulk formula in each ensemble member). These shortcomings should have been acknowledged in the paper.

One of the major uncertainties associated with both ocean reanalysis and objective analysis when considering OHC changes, is the sampling errors in the heterogeneous ocean observing network, which has evolved dramatically over the last few decades. E.g., the Southern Ocean is only properly sampled after Argo deployment since mid-2000. It is possible that the acceleration of a warming trend in global OHC after the 2000s is a result of better sampling (both horizontally and vertically) of the global ocean. This signal is visible in Fig.1, but was not fully discussed in this manuscript.

Thanks for the encouraging comments. We have addressed your main comments, adding some discussion about the issues raised by the reviewer. We fully agree that it is extremely difficult to account for all sources of uncertainties, and we focused on those that, from our previous experiences (cited papers in the manuscript), are the most important ones. We have added sentences and discussion in these regards, to acknowledge that accounting for all sources of uncertainty is difficult and our choices (e.g. same EN4 dataset but different corrections) may represent a shortcoming. This was added in section 4.3. Note we do not discuss all the items mentioned by the reviewer (for instance, we assume that uncertainty in volcanic events is implicit in the uncertainty in the atmospheric forcing, etc.). Also, we now refer to “major” instead of “all” sources of uncertainty (see also below).

Detailed remarks

Abstract

L8: Here, energy imbalance refers to the ocean's energy imbalance resulting from energy input/output through its boundaries (i.e., atmosphere, land, sea-ice, and ice shelf). It is not necessarily a result of human-induced climate change, as stated in this sentence.

Instead, we can point out that ocean heat content changes is a critical climate change indicator.

We were referring to the Earth's Energy Imbalance; we have rephrased the sentence following the reviewer's suggestions.

L13: “all sources of uncertainties”. This is a very strong statement, see my main remarks.

We fully agree, and changed it to "the major sources", also in all other occurrences in the manuscript.

L20-21: I suggest to clarify “observation procedure” and “surface data uncertainty” here.

Thanks. We changed to "observation calibration" (which is mostly the XBT correction procedures) and "sea surface temperature data uncertainty".

Ch. 1 Introduction

L33-37: I think this is a bit of an overstatement. It gives the impression that all changes in ocean heat content are solely due to anthropogenic causes, specifically the increase of greenhouse gases in the atmosphere induced by humans. However, other natural causes are associated with ocean heat content changes as well, namely volcanic and El Nino events.

We fully agree on this. However, what we meant is long-term changes (multi-decadal), which generally filter the internal variability typically of shorter time scale (e.g. <10 years, ENSO, volcanoes, etc.). We reformulated the sentence to account for this (adding the adjective “long-term”), and mention natural variability accordingly.

L39: due to the uniform radiative forcing “and surface wind variability”

Added, thanks

L44-48: Please clarify here what product are you referring to, objective analysis, ocean reanalysis, or real physical ocean?

Thanks, we rephrased completely the sentence to make it more clear and refer explicitly to pre-Argo period (lines 49-50).

L49-52: Uncertainties in reanalysis-based reconstructions can also arise from systematic model biases, not just vertical physics. I also suggest replacing "data assimilation configurations" with "data assimilation systems".

Corrected, thanks

L54-57: It's better to clarify that ocean reanalysis is not just the use of an OGCM to simulate the ocean state. In addition, it involves using observations to constrain the projected ocean state through various data assimilation methods.

Thanks, added (line 65). We indeed took it for granted and missed the important reference to data assimilation

L67: Please clarify what you mean by "all major sources of uncertainty" ?

We changed the sentence to "the major sources of uncertainty". The discussion is added in section 4.3

L68: For "objective analysis counterpart" add ref to Ch 4.2

Done

Ch. 2 Results

L81-82: please confirm that the global mean ocean warming rate of 0.43 +/- 0.08 W m⁻² is averaged over all earth surface (ocean/land/lakes) or just over sea surface ? "Earth's area" implies this value is averaged over all surface.

Correct, thanks. The caption of Table 1 explains this in more detail, so we prefer to leave the sentence unchanged to avoid redundant definitions.

L91: should be "anomalies" of global ocean heat content ...

Yes, corrected, thanks

L95-99: Do you know why the ENS-REA reanalysis show enhanced interannual variability with a sharper increase of positive trend after 2000? Is this something to do with Argo deployment since 2000s, which led to better sampling of the global ocean?

We have now provided a zoom for the 2005-onwards years in Figure 1 (subpanel, commented at lines 116-122), which shows consistent variability between ENS-REA and other (some independent) products. We have added comments accordingly.

L106-107: To me, the similarity between OHC trend/interannual variability between OA and ENS-ERA suggests that the OHC signal is dominated by the input observation data sets (EN4) and is less sensitive to the model background in your system.

Yes, we do agree on this. At the same time, when looking at the objective analysis system, reducing the frequency of corrections to monthly or changing the background affect the estimate. This indicates that most objective analyses show damped variability because of monthly frequency or choice of background. We have tried to better highlight this reasoning in the revised version of the manuscript (section 2.1).

L109-112: The Fig.1-middle panel shows that OA-MON and OA-SLS is more close to ENS-ERA (except after 2012), with the same sharper increases of positive trend after 2000. Only after 2012 OA-MON and OA-SLS signals are closer to GCOS20.

Exactly, thanks. The previous version of the paragraph did not touch on these details. Now, following yours and Reviewer 2's suggestions, we split the sentences to discuss the experiments individually.

L117: Any reason why larger interannual variability/trends in the Southern Ocean?

We added several references, as this point was also asked by Reviewer 1, which include a detailed explanation of the mechanisms. We have also added new panels in Figure 2 to stress that the enhanced interannual variability occurs in mesoscale active areas, with a detailed discussion (lines 142-149, 153-157, 161-163).

L120: Here "uncertainty" refers to 1 ensemble standard deviation. Uncertainty in Fig.1, however, refers to twice the ensemble std.

Thank you very much for spotting this inconsistency. We have redone the map plots to show twice the ensemble standard deviation, for consistency with the previous definition of uncertainty.

L119: Have you compared ocean heat content trend maps from ENS-REA (Fig.2) to that derived from CGOS20 data set? or other OA products ? Would be great if you can add a few sentences here.

The GCOS20 assessment is only global, and there are no products spanning the 1961-2022 period. We added the reference Johnson and Lyman (2020) and other publications and commented on the similarities.

L160-161: please clarify “our reanalysis system ... but is much steadier and more robust over time”.

We rephrased the sentence, we meant that GCOS uncertainty is quite variable while ours is not.

L593: running “mean” ensemble std ...?

The "running" (or "moving") operation is the standard deviation, so we prefer to leave the definition as it is.

Ch. 3 Discussion

L210: Not sure about the “unprecedented”. Suggest to rephrase.

Thanks. To our knowledge, there did not exist reanalyses with a larger number of members (SODA-historical, Giese et al. 2016 had 8 members; ORA reaches 10 in some versions; CERA-20C: 10 members; GREP 4 members). But while searching we found a CSIRO reanalysis CAFE60v1 with 96 members and similar resolution in the ocean. So, we removed the "unprecedented" adjective.

L213-214: See my main remarks. Ensemble generation method used in this study captures many but not all uncertainty sources (e.g. there is no sampling error in ocean observing network). I suggest to rephrase this sentence.

Not really clear this comment: the sentence at lines 213-21 of the originally submitted version of the manuscript refers to the use of a twin objective analysis system, not the ensemble generation/uncertainty. So, we have left the sentence unchanged. In any case, at the end of section 4.3 we discuss in detail the other possible sources of uncertainty not explicitly accounted for in our study.

L218: Again, not sure about “for the first time”. Also suggest to replace “the main sources” by “many major sources”.

Corrected: we removed "for the first time" and rephrased.

L227: please rephrase regarding to “all sources of uncertainty”.

We changed it to "The major sources..."

L237: Could you elaborate a bit on relationship between high uncertainties in the Tropics and El Niño events?

Sure, indeed this is not a novel result. See for instance Figure 7 in <https://link.springer.com/article/10.1007/s00382-018-4585-5>

"El Niño occurrences are characterized by an increase and broadening of vertical covariances with respect to the neutral years (especially in the eastern Tropical Pacific, see also Figure S3), due to the larger thermocline variability during El Niño compared to neutral and La Niña years." We added a sentence on this in the manuscript (lines 199-201).

Ch. 4 Methods

L311-314: please clarify on two sets of initial conditions used in the ENS-REA system.

The two initial conditions are taken valid for 1958 (beginning of the reanalysis, although the first three years are discarded as spinup) from a previous pilot reanalysis as initial conditions in 1948 and 1968. We added this explanation in the text, section 4 (lines 373-380).

REVIEWER COMMENTS

Reviewer #1 (Remarks to the Author):

The authors have adequately addressed my concerns and comments and I recommend the manuscript for publication.

I have identified a couple minor edits that the authors might want to look into.

-line 8: In my opinion it is the change in OHC that represents a climate change indicator. I recommend to rephrase to: "Long-term changes in OHC represent ... and are mostly"

-line 12: "understanding" instead of "understand".

-line 36: threaten?

-line 41: amount of

-line 46: "monitoring Earth's energy budget and climate change"

Reviewer #2 (Remarks to the Author):

I find most of my comments have been addressed in the revised version of the manuscript. But I have still 2 reservations about this manuscript

1) One of my comments has been addressed by the authors but it is not reflected in the revised version of the manuscript. It is my comment about the better results of recent reanalysis in simulating OHC changes compared to previous versions of the reanalysis. So far, Palmer et al. 2017 was the reference in terms of assessment of the OHC in reanalysis. Here the authors results are significantly better than in Palmer et al. 2007. The authors explained in the rebuttal why it was so. That is fine but these explanations do not appear in the revised manuscript. I think they should appear in a way or another (at least in a supplementary material) because they are important for the reader to understand what make the results better here than in Palmer et al. 2017. It will also give a touch on the actual progress of reanalysis

2) In the revised version of the manuscript the authors explain why their reanalysis-based estimate of the OHC shows more variability than the objective analysis one. They argue this is due to a limitations in objective analysis. Then why EBAF is also showing less interannual variability than the reanalysis? And CERES+ as well!. The author should answer this question or at least discuss it to convince the readership that the interannual signal shown in reanalysis is not spurious

I think that after correcting these 2 points the manuscript can be accepted for publication

I add hereafter a few minor comments

L45-46: "Ocean Heat Content 45 (OHC), representing the total amount heat stored in the ocean, is one of the most important indicators for ocean warming." This phrase is a tautology: ocean warming is evidently indicated by OHC changes. I suggest rephrasing

L82: "The present study aims to re-assess the OHC trends..." you probably mean the " OHC trends as estimated by reanalysis"

L112: Habuba->Hakuba

L127: I find the agreement is good with the geodetic solution but less good with the GCOS solution and not good with CERES. Can you comment on that? There is in general much more interannual variability in the OHU time series from reanalysis in situ data and geodetic estimate than in CERES. Why? Do you have hints on this? Note that the interannual variability in CERES+ is small as well. Why?

L132: What about the comparison with CERES?

L178: W.m-1 dec-1 -> W.m-2dec-1

Reviewer #3 (Remarks to the Author):

Most of my previous comments have been resolved to the satisfaction. However, further revision is needed. See my comment below.

One of my previous comment is about the uncertainty in OHC changes that is associated with ocean observing system sampling error (e.g. changes in spatial coverage of global ocean observing system, in particular before and after Argo). See my previous comment below

"One of the major uncertainties associated with both ocean reanalysis and objective analysis when considering OHC changes, is the sampling errors in the heterogeneous ocean observing network, which has evolved dramatically over the last few decades. E.g., the Southern Ocean is only properly sampled after Argo deployment since mid-2000. It is possible that the acceleration of a warming trend in global OHC after the 2000s is a result of better sampling (both horizontally and vertically) of the global ocean. This signal is visible in Fig.1, but was not fully discussed in this manuscript. "

Despite effort from the authors to address the high uncertainties OHC signals in the Southern Ocean, the

above point (of changing observing network) is not really addressed in the revised manuscript. I am still a bit suspicious about this increased trend after 2000, related to my comment above. This is encouraging to see the new sub-panel in Figure 1-a that includes comparison with CERES data. However, inter-annual variabilities from various products (ENS-REA and CERES related datasets) are really large. If you include also ZANNA19/ARANN/GCOS20 into this panel, how different are these datasets looks like?

In Fig.1-c panel, this increase of warming trend since 2000 is most pronounced in the Southern ET heat content anomaly, this is to me suggesting that the acceleration of warming trend may be a spatial sampling issue. A related question is, If Argo data is excluded in your DA system, do you have the same increase of positive trend in OHC anomalies after 2000 in global and in the Southern ET?

Response to the Reviewers' comments. 2nd round of reviews

Manuscript NCOMMS-23-08971 by Storto and Yang

Dear Reviewers,

First, we would like to thank you for the careful reading and the updated comments and suggestions. Below, we provide a point-by-point answer to your comments (reviewers in bold, our answer in normal font). Both authors agree with the changes. We have also corrected another couple of grammar issues within the manuscript. Additionally, in the revised version of the manuscript, we now refer to the updated name of our dataset as CIGAR (CIGAR: the Cnr Ismar Global historicAI Reanalysis), as data are released and advertised through a public website, and we have decided to update the name instead of the more generic and less appealing ENS-REA as before. Data and additional info are now released (see also <http://cigar.ismar.cnr.it>).

Reviewer #1 (Remarks to the Author):

The authors have adequately addressed my concerns and comments and I recommend the manuscript for publication. I have identified a couple minor edits that the authors might want to look into.

-line 8: In my opinion it is the change in OHC that represents a climate change indicator. I recommend to rephrase to: "Long-term changes in OHC represent ... and are mostly"

Corrected as suggested by the reviewer

-line 12: "understanding" instead of "understand".

Corrected as suggested by the reviewer

-line 36: threaten?

Corrected as suggested by the reviewer

-line 41: amount of

Corrected as suggested by the reviewer

-line 46: "monitoring Earth's energy budget and climate change"

Corrected as suggested by the reviewer

Reviewer #2 (Remarks to the Author):

I find most of my comments have been addressed in the revised version of the manuscript. But I have still 2 reservations about this manuscript

1) One of my comments has been addressed by the authors but it is not reflected in the revised version of the manuscript. It is my comment about the better results of recent reanalysis in simulating OHC changes compared to previous versions of the reanalysis. So far, Palmer et al. 2017 was the reference in terms of assessment of the OHC in reanalysis. Here the authors results are significantly better than in Palmer et al. 2007. The authors explained in the rebuttal why it was so. That is fine but these explanations do not appear in the revised manuscript. I think they should appear in a way or another (at least in a supplementary

material) because they are important for the reader to understand what make the results better here than in Palmer et al. 2017. It will also give a touch on the actual progress of reanalysis

Thanks a lot for this suggestion, we have been indeed hesitating whether to report part of the discussion of the previous response to the reviewers in the manuscript or not. The reviewer's suggestion pushed us to add this discussion, finally. We have now included a section on this in the Methods (*the new section 4.4*), which summarizes all the previous arguments.

2) In the revised version of the manuscript the authors explain why their reanalysis-based estimate of the OHC shows more variability than the objective analysis one. They argue this is due to a limitations in objective analysis. Then why EBAF is also showing less interannual variability than the reanalysis? And CERES+ as well!. The author should answer this question or at least discuss it to convince the readership that the interannual signal shown in reanalysis is not spurious

We thank the reviewer for raising this point; we believe, as demonstrated in previous works, that CERES-derived estimates contain high-frequency variability intrinsically different from that in the ocean heat uptake (and found smaller than the ocean heat uptake in many studies). For instance: i) Marti et al., 2022 (<https://essd.copernicus.org/articles/14/229/2022/>) removes signals lower than 3 years in the comparison with OHC tendencies; ii) Storto et al., 2022 (<https://doi.org/10.1175/JCLI-D-21-0726.1>) obtained coherence between the two datasets for the 3-year time-scale through cross-wavelet analysis; iii) Meyssignac et al., 2023, (<https://www.nature.com/articles/s41558-023-01735-z>) compares the EEI from the geodetic approach (inferred by altimetry minus gravimetry) to that of CERES after low-pass filtering the geodetic dataset, as this latter has more pronounced variability than CERES. This is because the signals at the top of the atmosphere respond to many atmospheric processes at very different scales (from tropical convection to decadal variability) which may be significantly different than the signals of the ocean heat uptake (see also the discussion in e.g. von Schuckmann et al., 2016, <https://doi.org/10.1038/nclimate2876>). As the effect of all these multi-scale processes on the different EEI estimates is known but far from being assessed in detail by the climate community, we prefer to discuss this issue in the revised version without being completely conclusive; indeed, it is still an open question which scales are fully comparable between the ocean heat uptake measured through the oceanic observing network and the EEI measured at the top of the atmosphere. We have shown and proved that the interannual variability from reanalyses is not spurious compared to objective analyses, but it is beyond the scope of the present work to show how TOA or ocean data manifest different frequency variability.

We have added a discussion on this point in section 2.1: *The high-frequency variability between CERES-based estimates and those from oceanic observations is known to be largely different, due to several weather and climate processes providing a different response in the TOA EEI compared to the ocean heat uptake (von Schuckmann et al., 2016). The climate community is converging toward comparing these two complementary datasets at frequencies slower than 3 years (Marti et al., 2022; Storto et al., 2022; Meyssignac et al., 2023). However, an accurate understanding of the scale coherence between the two is still an open question, whose answer is complicated by the relatively short temporal record of the data.*

I think that after correcting these 2 points the manuscript can be accepted for publication. I add hereafter a few minor comments. L45-46: "Ocean Heat Content 45 (OHC), representing the total amount heat stored in the ocean, is one of the most important indicators for ocean warming." This phrase is a tautology: ocean warming is evidently indicated by OHC changes. I suggest rephrasing

Corrected, thanks: "OHC... most important indicators of climate change in the ocean"

L82: "The present study aims to re-assess the OHC trends..." you probably mean the" OHC trends as estimated by reanalysis"

Corrected as suggested by the reviewer

L112: Habuba->Hakuba

Corrected

L127: I find the agreement is good with the geodetic solution but less good with the GCOS solution and not good with CERES. Can you comment on that? There is in general much more interannual variability in the OHU time series from reanalysis in situ data and geodetic estimate than in CERES. Why? Do you have hints on this? Note that the interannual variability in CERES+ is small as well. Why?

L132: What about the comparison with CERES?

Please see answer to the second major point above.

L178: W.m-1 dec-1 -> W.m-2dec-1

Corrected

Reviewer #3 (Remarks to the Author):

Most of my previous comments have been resolved to the satisfaction. However, further revision is needed. See my comment below. One of my previous comment is about the uncertainty in OHC changes that is associated with ocean observing system sampling error (e.g. changes in spatial coverage of global ocean observing system, in particular before and after Argo). See my previous comment below

"One of the major uncertainties associated with both ocean reanalysis and objective analysis when considering OHC changes, is the sampling errors in the heterogeneous ocean observing network, which has evolved dramatically over the last few decades. E.g., the Southern Ocean is only properly sampled after Argo deployment since mid-2000. It is possible that the acceleration of a warming trend in global OHC after the 2000s is a result of better sampling (both horizontally and vertically) of the global ocean. This signal is visible in Fig.1, but was not fully discussed in this manuscript. " Despite effort from the authors to address the high uncertainties OHC signals in the Southern Ocean, the above point (of changing observing network) is not really addressed in the revised manuscript. I am still a bit suspicious about this increased trend after 2000, related to my comment above. This is encouraging to see the new sub-panel in Figure 1-a that includes comparison with CERES data. However, inter-annual variabilities from various products (ENS-REA and CERES related datasets) are really large. If you include also ZANNA19/ARANN/GCOS20 into this panel, how different are these datasets looks like? In Fig.1-c panel, this increase of warming trend since 2000 is most pronounced in the Southern ET heat content anomaly, this is to me suggesting that the acceleration of warming trend may be a spatial sampling issue. A related question is, If Argo data is excluded in your DA system, do you have the same increase of positive trend in OHC anomalies after 2000 in global and in the Southern ET?

We thank the reviewer for clarifying this point, and we split the reply into two parts, one regarding the interannual variability, and one about the observational sampling

i) Interannual variability

Regarding the interannual variability, Figure 1a already includes GCOS20, and GCOS variability is very well aligned with that of ENS-REA (now called CIGAR). The only datasets showing damped variability are those related to CERES, and we argue that this is because CERES data measure a quantity intrinsically different from the OHC assessed by reanalyses or objective analyses, as it refers to the TOA EEI. For a detailed discussion, please see the answer to Reviewer 2, point 2). We have added a paragraph in section 2.1 to explain in detail

the reasoning, corroborated by many studies in the recent literature, which suggest comparing the two datasets at a 3-year timescale, as timescales shorter than that respond to a different range of processes between OHC and TOA.

ii) Observational sampling

Regarding the possible impact of the observational sampling on the estimates (global, or southern ocean), this is certainly an important point to discuss. We want to stress that in the second revision, we have already added (lines 101-105) a quantitative comparison of the OHC acceleration with independent estimates from Hakuba et al. (2021) (using TOA measurements), which already indicates the consistency of the two estimates. This in turn means that observational sampling is not critical in our estimates, and this is also indirectly confirmed by the bottom panel of Figure 4 in the manuscript (Figure 4c).

Another way to look at this is by withholding a large fraction of Argo data from the OA experiments (doing this in reanalysis experiments is too computationally demanding, therefore we use OA as a benchmark, provided that we have already demonstrated that the two have comparable long-term warming/acceleration signals).

The figure and table on the next page summarize the results at the global scale for the full period 1961-2022, with regional variations being negligible as well. There is no impact (statistically) of withholding a variable number of Argo floats: OA-50%, OA-75%, and OA-90% refer to the experiments where 50%, 75%, and 90%, respectively, of Argo floats have been randomly withheld, while OA-1000m when Argo float measurements below 1000m of depth have been rejected, to mimic the case where only upper ocean measurements are assimilated. Differences in terms of metrics of warming, acceleration, and interannual variability are very close to each other and indicate that the observational sampling has a negligible impact on the resulting warming and acceleration. We do not consider the case of 100% Argo rejection, which is not realistic considering that XBT and CTD data sampling decreased from the mid-1990s (see e.g. <https://www.aoml.noaa.gov/phod/goos/xbtscience/data.php>; Abraham et al., 2013, <https://doi.org/10.1002/rog.20022>; Meyssignac et al., 2019, <https://doi.org/10.3389/fmars.2019.00432>; Goes et al., 2020, <https://doi.org/10.1175/JTECH-D-20-0027.1>).

We have now mentioned this result while commenting on Figure 4c (lines 217-220 of the revised manuscript), but since the results of these new tests do not differ from OA, we think there is no need to include explicitly in the figures of the manuscript these new results.

Heat content anomaly Global

Figure R1. Global OHC anomaly for the experiments presented in the response above.

Experiment	Trend	Acceleration	Interannual Variability
OA	0.42	0.12	0.18
OA-50%	0.41	0.12	0.17
OA-75%	0.40	0.11	0.17
OA-90%	0.41	0.12	0.18
OA-1000m	0.39	0.10	0.16

Table R1. Reporting trend, acceleration, and interannual variability (as defined in the manuscript) for the additional OA experiments presented above. Note, values are like in Table 1 of the manuscript, except the period here is the full 1961-2022 period (therefore numbers may vary slightly).

REVIEWERS' COMMENTS

Reviewer #3 (Remarks to the Author):

Thanks to the authors who takes time to resolve my concerns about uncertainties in OHC trend related with observation sampling errors. And I welcome the effort that the authors have made to address this issue, especially new content dedicated to discuss this topic (between L208-223). I recommend this manuscript for publication.

One minor comment

We need to be careful when discuss OA experiments by withholding a fraction of Argo data because there is no Argo data before 2000. This approach (of withholding Argo) can be to sample uncertainties in OA reconstruction of OHC trend in the Argo period, but can not tell much about observation sampling error in the Pre-Argo period.